# GENERALIZED SPHERICAL NEURAL OPERATORS: GREEN'S FUNCTION FORMULATION

**Hao Tang**[1]     **Hao Chen**[2]     **Chao Li**[1,2]

[1]University of Dundee, United Kingdom
[2]University of Cambridge, United Kingdom

## ABSTRACT

Neural operators offer powerful approaches for solving parametric partial differential equations, but extending them to spherical domains remains challenging due to the need to preserve intrinsic geometry while avoiding distortions that break rotational consistency. Existing spherical operators rely on rotational equivariance but often lack the flexibility for real-world complexity. We propose a generalized operator-design framework based on **designable Green's function** and its harmonic expansion, establishing a solid operator-theoretic foundation for spherical learning. Based on this, we propose an absolute and relative position-dependent Green's function that enables flexible balance of equivariance and invariance for real-world modeling. The resulting operator, *Green's-function Spherical Neural Operator* (GSNO) with a novel spectral learning method, can adapt to non-equivariant systems while retaining spherical geometry, spectral efficiency and grid invariance. To exploit GSNO, we develop SHNet, a hierarchical architecture that combines multi-scale spectral modeling with spherical up-down sampling, enhancing global feature representation. Evaluations on diffusion MRI, shallow water dynamics, and global weather forecasting, GSNO and SHNet consistently outperform state-of-the-art methods. The theoretical and experimental results position GSNO as a principled and generalized framework for spherical operator design and learning, bridging rigorous theory with real-world complexity. The code is available at: `https://github.com/haot2025/GSNO`.

## 1 INTRODUCTION

**Background:** Solving parameterized partial differential equations (PDEs) is a fundamental task across science and engineering. Applications such as weather forecasting, fluid dynamics, and neuroimaging often involve high-order PDEs whose numerical solutions are computationally expensive and intractable. Emerging neural operators provide a promising alternative by approximating solution operators directly from data (Kovachki et al., 2024). Earlier approaches (Lu et al., 2019; 2021; Bhattacharya et al., 2021) learned mapping between function spaces using neural networks but struggled to scale to high-dimensional PDEs. To address this, the Fourier Neural Operator (FNO) (Li et al., 2020a) leverages Fast Fourier transforms (FFTs) to learn in the frequency domain, capturing global patterns and high-frequency modes (Kovachki et al., 2024). This inspired a new generation of operator learning methods and advanced large-scale applications such as high-resolution climate prediction (Pathak et al., 2022; Liu et al., 2024).

However, FNOs rely on the standard Fourier transform and assume Euclidean geometry. On non-Euclidean manifolds such as the sphere (Bonev et al., 2023), FFT-based representations introduce distortions: small polar displacements can map to large Cartesian displacements, breaking spatial coherence and degrading performance. To address this, Spherical Fourier Neural Operator (SFNO) is proposed (Bonev et al., 2023), replacing the FFT with the Spherical Harmonic Transform (SHT). By projecting functions onto spherical harmonic bases, SFNO preserves rotational equivariance on the sphere, ensuring stability under arbitrary input rotations. SFNO-based methods have achieved strong performance on some spherical tasks, *e.g.*, weather prediction (Lin et al., 2023; Mahesh et al., 2024a;b; Hu et al., 2025).

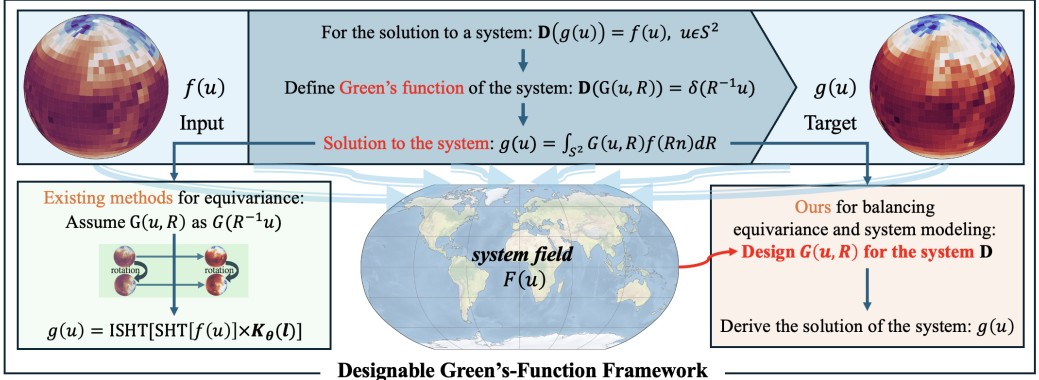

Figure 1: Method comparison under the Green's function framework (Details in Sec. 4.1). (Left) Existing approaches use SHT-based spectral learning (symmetric kernel). (Right) Our method extends sysyem characteristics by designable Green's function to derive the system-constraint solution.

Despite these advances, fundamental challenges yet to be addressed for spherical operator learning:

**(1)** Theory: most spherical neural operators are rigorously constructed by Spherical Harmonic Transform and spherical convolution theorem, thereby extending the FNO to the sphere, rather than derived from the integral solution of sphere-native PDEs (Bonev et al., 2023; Lin et al., 2023; Mahesh et al., 2024a;b; Hu et al., 2025). This lack of rigorous grounding obscures the physical meaning of the learned integral kernels and limits their generalization and extension in physical systems. **(2)** Neural Operator Block: Physical systems involve non-equivariance and asymmetric constraints, such as boundary effects, position-relevant patterns, or heterogeneous media (Ye et al., 2022; 2023; Lucarini & Chekroun, 2024; Behroozi et al., 2025), which cannot be captured if operators depend solely on equivariance. However, by underlying design, most spherical operators enforce strict rotational equivariance to gain efficiency for spectral learning, overlooking such asymmetry and thereby limiting their ability to model complex, high-order nonlinear phenomena. **(3)** Neural Operator Network: Existing spherical operator networks use ResNet-style architecture Bonev et al. (2023) operating at a single scale to capture global structure, but may limit the expressiveness in modeling climate dynamics (Zhao et al., 2019; 2021; Hu et al., 2025).

**Our Approach:** **(1)** We first derive the principled formulation of spherical operator learning that begins from a designable Green's function. This derivation extends the Green's function naturally from the sphere to the spherical harmonic domain, providing **a generalized operator-theoretic framework** of spherical learning, which supports the simulation of different complex systems by designing different Green's functions. **(2)** Based on this foundation, we enhance the flexibility of spherical operators on real-world systems by designing an absolute and relative position-dependent Green's function as an explicit inductive bias, deriving the corresponding response to explicitly balance equivariance and invariance. The yielded Green's-Function Spherical Neural Operator (GSNO) balances the spectral efficiency with the capacity to capture complex system constraints g boundaries and distortions while retaining spherical geometry and grid invariance, enabling the nonlinear modeling in real-world, non-equivariant physical systems. **(3)** Finally, we propose the Spherical Harmonic Neural Operator Network (SHNet), a hierarchical architecture that integrates GSNO with multi-scale spectral modeling through spherical up–down sampling, enhancing the expressive power while preserving geometric consistency across resolutions.

**Contributions:** This work mainly makes the following contributions:

1. **A generalized theoretical framework** to derive system-based operator solutions based on the **designable spherical Green's functions** (Sec. 4.1).

2. We design an absolute and relative position-dependent Green's function and derive the corresponding response to propose a novel operator (GSNO) that flexibly balances equivariance and invariance and models complex systems while retaining grid invariance(Sec. 4.2).

3. We design SHNet, a multi-scale spherical network based on GSNO (Sec. 4.3).

4. GSNO and SHNet are evaluated on the diffusion MRI modeling of brain microstructure, spherical shallow water equations and weather prediction, demonstrating consistent improvements in performance over other state-of-the-art models.

## 2 RELATED WORK

**Geometric equivariance and complex constraints.** Geometric equivariance has been explored through transformations such as translation, scaling, and dilation of filters, in both discrete and continuous domains (Xu et al., 2014; Sosnovik et al., 2021; Rahman & Yeh, 2023; Chen et al., 2023). These approaches contribute to the framework of geometric deep learning by incorporating rich symmetry groups (Cohen & Welling, 2016; Bronstein et al., 2021). Several methods embed inductive biases to enhance generalization (Wad et al., 2022; Han et al., 2022); for instance, CNNs are inherently equivariant to translations (Li et al., 2021). Group-equivariant neural networks (Cohen & Welling, 2016; Cohen et al., 2018), along with spectral approaches such as DISCO (Ocampo et al., 2022), further exploit symmetries to improve learning efficacy. However, for real-world modeling, there are some components that relax rotational equivariance in most spherical CNN-based models, *e.g.*, residual pathways, position embeddings, activation functions and local operations (Cohen & Welling, 2016; Cohen et al., 2018; Bonev et al., 2023; Finzi et al., 2021; Liu-Schiaffini et al., 2024). Other methods are proposed to further introduce complex constraints and relax strict equivariance for better real-world modeling capabilities (Finzi et al., 2021; Huang et al., 2022; Wang et al., 2022; Duval et al., 2023; Pertigkiozoglou et al., 2024; Zheng et al., 2024; Li et al., 2024).

**Neural operators with multi-scale modeling.** Recent extensions introduce multi-scale learning to enhance feature representation. Some neural operators (Li et al., 2020c; Lütjens et al., 2022; Raonic et al., 2023; You et al., 2024; Liang et al., 2024; Liu et al., 2025) tackle this by combining both upsampling and downsampling operations capturing multiscale information. However, the operations may introduce aliasing artifacts (Ronneberger et al., 2015; Karras et al., 2021) and distortions, limiting modeling accuracy, especially in spherical space (Zhao et al., 2019; 2021).

## 3 PRELIMINARY

### 3.1 SPHERICAL HARMONIC

The spherical harmonic function (Müller, 2006) $Y_l^m(\theta, \phi)$, with integer degrees $l \geq 0$ and orders $|m| \leq l$, forms an orthonormal basis for square-integrable functions on the unit sphere $S^2$. Spherical harmonic transform (SHT) decomposes a function $f \in L^2(S^2)$ into its harmonic coefficients:

$$\text{SHT}[f](l, m) = \int_{S^2} f(\omega) \overline{Y_l^m(\omega)} \, d\omega, \quad \omega \in S^2 \tag{1}$$

The orginal function $f$ is exactly reconstructed via inverse spherical harmonic transform (ISHT):

$$f(\omega) = \text{ISHT}(\text{SHT}[f](l, m)) = \sum_{l=0}^{\infty} \sum_{m=-l}^{l} \text{SHT}[f](l, m) Y_l^m(\omega) \tag{2}$$

Given two functions $f$ and $h$ defined on the sphere $S^2$, their spherical convolution is defined as (Driscoll & Healy, 1994):

$$(f * h)(\omega) = \int_{SO(3)} f(Rn) h(R^{-1}\omega) \, dR \tag{3}$$

where $n \in S^2$ denotes the north pole, and $R$ is an element of the three-dimensional rotation group $SO(3)$. The SHT coefficients are given by:

$$\text{SHT}[(f * h)](l, m) = \int_{S^2} (f * h)(\omega) \overline{Y_l^m(\omega)} \, d\omega \tag{4}$$

## 3.2 SPHERICAL CONVOLUTION THEOREM

To further derive the SHT coefficients of a spherical convolution, the spherical convolution theorem (Driscoll & Healy, 1994) is applied as follows:

$$\text{SHT}[(f * h)](l, m) = 2\pi \sqrt{\frac{4\pi}{2l + 1}} \cdot \text{SHT}[h](l, 0) \cdot \text{SHT}[f](l, m). \tag{5}$$

This result demonstrates that spherical convolution in the harmonic domain corresponds to a product of the SHT coefficients of $f$ and $h$ (with $h$ restricted to the $m = 0$ mode) in the frequency domain. The full derivation is in the Appendix A. By replacing the classical convolution theorem with this spherical convolution theorem, SFNOs achieve frequency-domain parameterization on the sphere (Bonev et al., 2023; Lin et al., 2023; Hu et al., 2025). To enhance interpretability, we introduce two complementary SHT-based neural operator derivations in spherical space (Sec. 4.1).

## 4 METHODOLOGY

### 4.1 OPERATOR FRAMEWORK VIA SPHERICAL GREEN'S FUNCTION

The Green's function method offers a classical strategy to solve PDEs, where the solution is expressed as a convolution integral with the Green's function as the kernel (Li et al., 2020b). Differently, we define $D$, a linear differential operator on the sphere, and consider the following PDE:

$$D(g(u)) = f(u), \quad u \in S^2, \tag{6}$$

where $f(u)$ is the input function and $g(u)$ is the target solution. The proposed spherical Green's function $G$, associated with $D$, is defined by the property:

$$D(G(u, R)) = \delta(R^{-1}u) = \begin{cases} \infty, & Rn = u, \\ 0, & Rn \neq u, \end{cases} \tag{7}$$

where $\delta(.)$ denoting the Dirac delta function defined on the sphere, and $R \in SO(3)$ represents a rotation from the north pole $n$. Using the Green function, the solution to Equation 6 is:

$$g(u) = \int_{S^2} G(u, R) f(Rn) \, dR. \tag{8}$$

To verify this solution, we apply the operator $D$ to both sides of Equation 8:

$$\begin{aligned} D(g(u)) &= D\left(\int_{S^2} G(u, R) f(Rn) \, dR\right) \\ &= \int_{S^2} D(G(u, R)) f(Rn) \, dR \\ &= \int_{S^2} \delta(R^{-1}u) f(Rn) \, dR \\ &= f(u). \end{aligned} \tag{9}$$

In this framework, we propose an operator design method based on the **designable Green's function** to simulate different systems. For example, classic spectral operators assume the strict rotational equivariant mapping/system: Design $G(u, R)$ as $G(R^{-1}u)$. Under this relative-position dependent Green's function $G(R^{-1}u)$, the prediction target $g(u)$ is given by:

$$g(u) = \int_{S^2} G(R^{-1}u) f(Rn) \, dR. \tag{10}$$

Applying the spherical convolution theorem (Equation 5), the spherical harmonic transform of $g(u)$ is simplified as:

$$\begin{aligned} \text{SHT}[g(u)](l, m) &= \text{SHT}[(f * G)](l, m) \\ &= 2\pi \sqrt{\frac{4\pi}{2l + 1}} \cdot \text{SHT}[G](l, 0) \cdot \text{SHT}[f](l, m). \end{aligned} \tag{11}$$

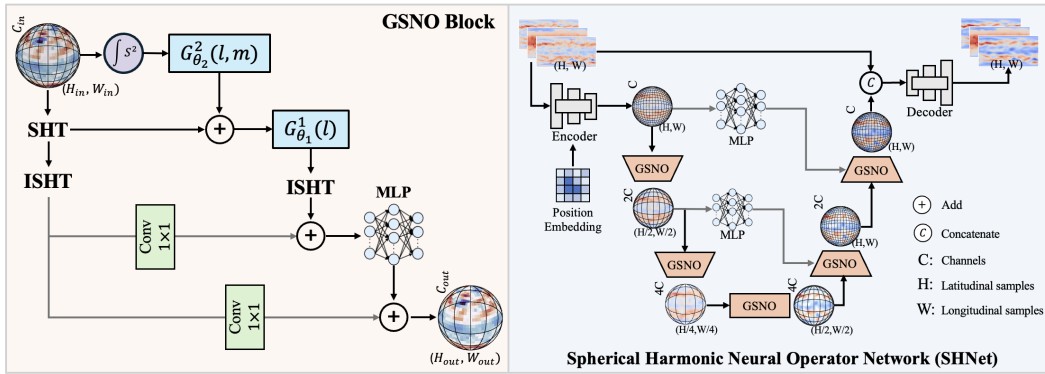

Figure 2: The proposed GSNO block (left) and the architecture of SHNet (right). SHT and ISHT represent spherical harmonic transformation and inverse transformation. Multi-layer perceptrons (MLPs) and two convolutional layers are used for channel interaction.

Thus, the target function $g(u)$ is reconstructed via the ISHT (Equation 2):

$$g(u) = \text{ISHT}(G_\theta(l) \cdot \text{SHT}[f](l, m)), \tag{12}$$

where $G_\theta(l)$ denote the learnable spectral weights parameterized by the neural operator. This consistency with SFNOs verifies the feasibility of this **designable Green's-function framework**.

**Spherical Harmonic Extension.** The above derivation relies on spherical convolution theorem. To extend, we propose to parse neural operators entirely from spherical harmonics and present a derivation based on the harmonic extension of Green's function (Groemer, 1996). Green's function is extended as:

$$G(u) = \sum_{l=0}^{\infty} \sum_{m=-l}^{l} \text{SHT}[G](l, m) Y_l^m(u). \tag{13}$$

Therefore, the designable form of the Green's function $G(u, R)$ is naturally extended to harmonic domain without disrupting spherical geometry. Substituting into the definition of $g(u)$, we derive the SHT of target as follows:

$$
\begin{aligned}
\text{SHT}[g(u)](l, m) &= \int_{SO(3)} f(Rn) \left( \int_{S^2} G(R^{-1}u) \overline{Y_l^m(u)} \, du \right) dR \\
&= \int_{SO(3)} f(Rn) \left( \int_{S^2} \sum_{|m'| \le l'} \text{SHT}[G](l', m') Y_{l'}^{m'}(R^{-1}u) \overline{Y_l^m(u)} \, du \right) dR \\
&= 2\pi \sqrt{\frac{4\pi}{2l+1}} \cdot \text{SHT}[G](l, 0) \cdot \text{SHT}[f](l, m).
\end{aligned}
\tag{14}
$$

Key derivation steps are detailed in Appendix B.1. Note that the derived result is also consistent with the outcome obtained from Equation 11 (the target ($g(u)$) follows Equation 12).

Therefore, this section presents a comprehensive and solid theoretical framework to simulate different systems based on **designable Green's functions**, deriving the corresponding operator solutions, shown in the Figure 1. Equation (11) and (14) are both the solution derived from the relative-position dependent Green's function, demonstrating the designable and scalable forms of Green's function: $G(R^{-1}u)$ and its spherical harmonic extension.

## 4.2 GREEN'S-FUNCTION SPHERICAL NEURAL OPERATOR

Spherical spectral convolutions assume the Green's function depends solely on the relative position $R^{-1}u$ through rotational equivariance ($G(R, u) = G(R^{-1}u)$). However, this strong assumption ignores the applicability to real-world non-equivariance, *e.g.*, position-dependent geophysical settings

characterized by anisotropy, local heterogeneity, and non-periodic boundary conditions (Van Essen et al., 2013; Lucarini & Chekroun, 2024). To address this limitation, the Green's function is designed by combining the relative and absolute position-dependent terms. This enables modeling of invariant physical properties without sacrificing the efficiency of spectral-domain computation. We define the extended Green's function hypothesis as:

$$
\begin{aligned}
G(R, u) &= \sum_{l',m'} \text{SHT}[G_1]_{l'}^{m'} [Y_{l'}^{m'}(R^{-1}u) + bias(u)] \\
&= \sum_{l',m'} \text{SHT}[G_1]_{l'}^{m'} [Y_{l'}^{m'}(R^{-1}u) + \text{SHT}[G_2]_{l'}^{m'} Y_{l'}^{m'}(u)] \\
&= \underbrace{\sum_{l',m'} \text{SHT}[G_1]_{l'}^{m'} Y_{l'}^{m'}(R^{-1}u)}_{\text{Original Term } (\mathcal{T}_{\text{orig}})} + \underbrace{\sum_{l',m'} \text{SHT}[G_1]_{l'}^{m'} \text{SHT}[G_2]_{l'}^{m'} Y_{l'}^{m'}(u)}_{\text{Correction Term } (\mathcal{T}_{\text{corr}})}.
\end{aligned}
\tag{15}
$$

The total operator thus comprises two components: $\mathcal{T}_{\text{orig}}$ is the original equivariant term, preserving the rotational equivariant characteristics, while $\mathcal{T}_{\text{corr}}$, a novel, learnable, non-equivariant term that captures spatial constraints and heterogeneities, models complex constraint conditions. Based on this formulation, the SHT result of target $g(u)$ is:

$$
\text{SHT}[g(u)](l, m) = \underbrace{\text{SHT}[(f * \mathcal{T}_{\text{orig}})](l, m)}_{\text{I}(\mathcal{T}_{\text{orig}})} + \underbrace{\text{SHT}[(f * \mathcal{T}_{\text{corr}})](l, m)}_{\text{I}(\mathcal{T}_{\text{corr}})}.
\tag{16}
$$

Following the derivation in Sec. 4.1, the equivariant component is:

$$
\begin{aligned}
\text{I}(\mathcal{T}_{\text{orig}}) &= 2\pi \sqrt{\frac{4\pi}{2l+1}} \cdot \text{SHT}[G_1](l, 0) \cdot \text{SHT}[f](l, m) \\
&= G_{\theta_1}^1(l) \cdot \text{SHT}[f](l, m).
\end{aligned}
\tag{17}
$$

For the correction term:

$$
\begin{aligned}
\text{I}(\mathcal{T}_{\text{corr}}) &= \int_{SO(3)} f(Rn) \left( \int_{S^2} \left[ \sum_{l',m'} \text{SHT}[G_1]_{l'}^{m'} \text{SHT}[G_2]_{l'}^{m'} Y_{l'}^{m'}(u) \right] \overline{Y_l^m(u)} \, du \right) dR \\
&= \text{SHT}[G_1](l, m) \text{SHT}[G_2](l, m) \int_{SO(3)} f(Rn) dR \\
&= G_{\theta_1}^1(l, m) \cdot C_f \cdot G_{\theta_2}^2(l, m),
\end{aligned}
\tag{18}
$$

where $C_f$ is the spherical integral of the input function ($f(u)$). The original term $\text{I}(\mathcal{T}_{\text{orig}})$ and the correction term $\text{I}(\mathcal{T}_{\text{corr}})$ share $G_{\theta_1}^1(l)$. To reduce parameter redundancy between $G_{\theta_1}^1(l, m)$ and $G_{\theta_2}^2(l, m)$ and retain high computational efficiency, we simplify Equation 18 to:

$$
\text{I}(\mathcal{T}_{\text{corr}}) = G_{\theta_1}^1(l) \cdot C_f \cdot G_{\theta_2}^2(l, m).
\tag{19}
$$

Combining both components $\text{I}(\mathcal{T}_{\text{orig}})$ and $\text{I}(\mathcal{T}_{\text{corr}})$, the final output is:

$$
\begin{aligned}
g(u) &= \text{ISHT}[\text{I}(\mathcal{T}_{\text{orig}}) + \text{I}(\mathcal{T}_{\text{corr}})] \\
&= \text{ISHT}[G_{\theta_1}^1(l) \cdot (\text{SHT}[f](l, m) + C_f \cdot G_{\theta_2}^2(l, m))].
\end{aligned}
\tag{20}
$$

The parameter quantity of $G_{\theta_1}^1(l)$ is much larger than that of $G_{\theta_2}^2(l, m)$ in applications because $G_{\theta_1}^1(l)$, as the outer main weight, needs to represent cross-channel interaction, while $G_{\theta_2}^2(l, m)$, as the inner biased weight of the same shape as $\text{SHT}[f](l, m)$, has no interaction requirement with $C_f$.

Based on Equation 20, GSNO block is designed (Figure 2, left). The input spherical feature $f$ is first transformed into spherical harmonic coefficients through SHT. In parallel, the spherical integral $C_f$ of input $f$ is used to modulate the kernel $G_{\theta_2}^2(l, m)$ to obtain the complete correction term. Then, the sum of the spherical harmonic coefficient and the correction term undergo a generalized multiplication of the tensor contraction with $G_{\theta_1}^1(l)$ to obtain the transformed spherical harmonic

Table 1: MRE↓ ($\times 10^{-3}$) on SSWE of 3 test variables and their means at 5h and 10h across models. Bold indicates best performance. H, V and D are variables on SSWE tasks.

| Method | MRE at 5 h | | | | MRE at 10 h | | | |
|---|---|---|---|---|---|---|---|---|
| | H | V | D | Average | H | V | D | Average |
| ClimaX (Nguyen et al., 2023) | $5.27 \pm 0.16$ | $222 \pm 9.00$ | $1000 \pm 0.13$ | $409 \pm 3.00$ | $5.87 \pm 0.19$ | $324 \pm 11.7$ | $1000 \pm 0.18$ | $443 \pm 3.90$ |
| FourCastNet (Pathak et al., 2022) | $3.40 \pm 0.12$ | $187 \pm 4.86$ | $716 \pm 7.69$ | $302 \pm 2.99$ | $3.56 \pm 0.08$ | $319 \pm 6.15$ | $737 \pm 8.95$ | $353 \pm 3.45$ |
| SFNONet (Bonev et al., 2023) | $1.39 \pm 0.07$ | $145 \pm 11.1$ | $295 \pm 8.87$ | $147 \pm 4.89$ | $1.68 \pm 0.12$ | $229 \pm 15.6$ | $353 \pm 13.5$ | $195 \pm 6.86$ |
| SHNet (Ours) | $\mathbf{1.26 \pm 0.07}$ | $\mathbf{134 \pm 9.30}$ | $\mathbf{273 \pm 5.91}$ | $\mathbf{136 \pm 3.73}$ | $\mathbf{1.49 \pm 0.09}$ | $\mathbf{201 \pm 11.9}$ | $\mathbf{326 \pm 12.1}$ | $\mathbf{176 \pm 5.71}$ |

coefficient, which is finally converted to the spherical characteristics of the transformed output $g$ through ISHT. Thus, the complex transformations and parameter learning in the spherical space are transformed into simple operations in the frequency domain. To further enhance nonlinear modeling, we apply a multi-layer perceptron (MLP) with two $1 \times 1$ convolutional layers and GELU activation function for channel interaction. Two additional light-weight convolutional layers are used for linear interaction, channel transformation, and skip connections (residuals).

GSNO combines equivariant response ($G_1$) and invariant response ($G_2$), enabling simultaneous modeling of real non-equivariant systems. Specifically, $G_1$ encodes more dynamic features, $G_2$ provides mode-level spectral embedding, which explicitly encodes systematic and more stable constraints, including local non-uniformity and boundary constraints (*e.g.*,weather prediction affected by terrain) while retaining grid invariance and avoiding influence on the feature representation relying on rotational equivariance by sufficient parameter learning. In addition, GSNO maintains high computational efficiency and low parameters as SFNO. In summary, GSNO improves spherical operators by relaxing the strict $SO(3)$ equivariance flexibly, allowing real-world modeling without disrupting spherical geometry.

### 4.3 SPHERICAL HARMONIC NEURAL OPERATOR NETWORK

While spectral convolution excels in capturing global dependencies, it is limited in multi-scale modeling. Building on GSNO, we propose a multi-scale spherical harmonic network, SHNet (Figure 2 right). SHNet adopts a U-Net structure (Zhao et al., 2019; 2021; Hu et al., 2025) for expansion and compression of spatial and channel, and incorporates position embedding to enhance global modeling Bonev et al. (2023). As a core component, the GSNO block performs scale transformation of the spatial and spectral domains based on SHT and ISHT, and channel transformation via MLP. Specifically, we achieve scale transformation by modifying the number of sampling points along $\theta$ (latitude) and $\phi$ (longitude) in Equation 2 and the degree $l$ in Equation 1, which avoids distortions caused by traditional up- and down-sampling. The downsampling blocks reduce the sampling points and enhance feature expression through MLP, thereby realizing the abstraction of large-scale features. The upsampling restores higher-frequency content, with skip connections providing direct access to high-resolution information. This desgin allows SHNet to capture multi-scale interaction via geometric up- and down-sampling on the sphere (details in Appendix B.2).

## 5 EXPERIMENTS

We compare GSNO and SHNet to other state-of-the-art methods under identical experimental setup and configuration (Liu et al., 2024), *e.g.*, Transformer-based ClimaX (Nguyen et al., 2023), FNO-based FourCastNet (Pathak et al., 2022), and SFNO-based SFNONet (Bonev et al., 2023) (detailed in Appendix C.1). The models are compared on diffusion magnetic resonance imaging (dMRI) modeling and two autoregressive spherical datasets. We also conduct ablation experiments to further verify the effectiveness of the proposed GSNO block and SHNet structure. All experiments are implemented through PyTorch on 16GB A5000 GPUs.

### 5.1 SPHERICAL SHALLOW WATER EQUATIONS

Spherical Shallow Water Equations (SSWE) form a nonlinear hyperbolic PDEs system that model the motion of thin-layer fluids on a rotating sphere. The core underlying assumption is the shallow water approximation, where the vertical scale of the fluid layer is much smaller than the horizontal scale (Bonev et al., 2018). Following (Bonev et al., 2018; 2023), we generate the SSWE simulation

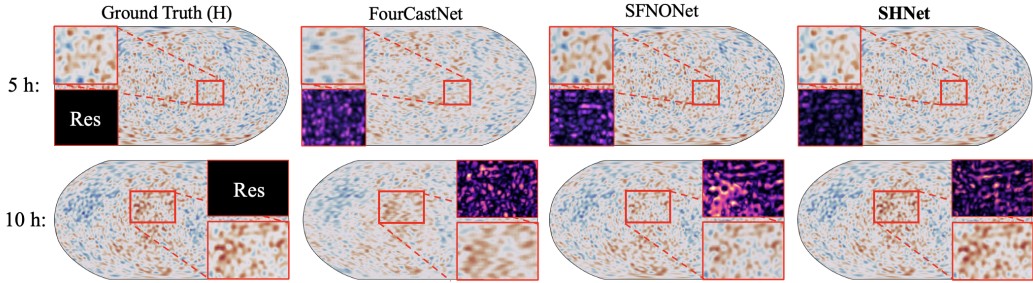

Figure 3: Predicted geopotential height (H) at 5h and 10h from different methods. Zoom in and provide the absolute residual (Res) to the ground truth for clarity (darker is better).

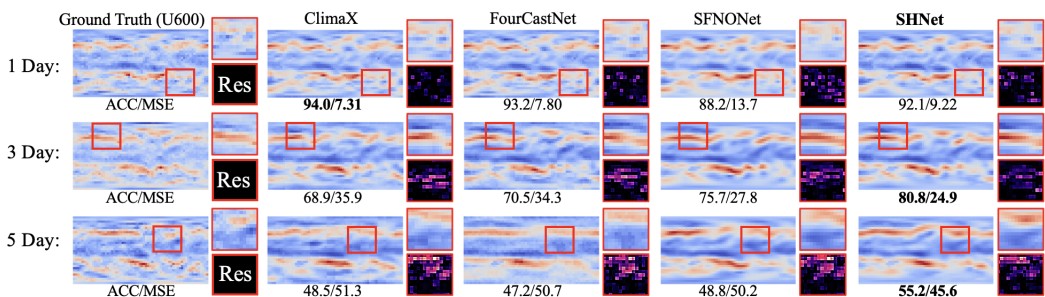

Figure 4: Predicted V600 samples of different methods. Each image set shows the main prediction (left), a zoomed-in region (top-right), and the absolute residual (Res) to the ground truth (bottom-right, darker is better). The performance is evaluated using ACC($\times 10^{-2}$)/MSE ($\times 10^{-2}$).

using a classical spectral solver (Giraldo, 2001). This dataset is well-suited to verify our model due to its spherical geometric characteristics (Appendix C.2, spatial resolution of $256 \times 512$, time step of 60s, 3 channel dimensions: geopotential height (H), vorticity (V), and divergence (D)). All comparisons use the identical dataset and setting: 50 epochs containing 256 samples each, batch size of 16, Adam optimizer with learning rate of $1 \times 10^{-3}$, and spherical weighted mean relative loss (details in Appendix C.1). Models are tested on the dataset generated from 50 initial conditions (50 samples) and evaluated using MRE for each variable. The results (Table 1) show that our method achieves the best performance on all variables and time scales, where the average increase is 7.5% at 5h and 9.7% at 10h. Besides, we conduct additional verifications under the exact same setting as SFNONet and at different resolution ($128 \times 256$) for reference, the results are summarized in the Table 6 and 7 in Appendix. For the predicted geopotential height (Figure 3), SFNONet and FourCastNet show more errors in the zoomed-in region, compared to our model. ClmaX is not presented due to its poor results.

## 5.2 WEATHER FORECASTING

To further assess the performance of our methods, we utilize WeatherBench (Rasp et al., 2020), a widely used autoregressive global weather prediction benchmark that offers data at multiple spatial resolutions. We select the dataset with a spatial resolution of $5.625°$ ($32 \times 64$) and a temporal resolution of 1 hour for evaluation. The training period spans 1979-2015, validation is conducted on data from 2016, and testing covers 2017-2018. The full dataset includes 24 meteorological variables, and we select six key variables as prediction targets (Liu et al., 2024) (details in Appendix C.3). All models are under the same settings: batch size 128, 50 epochs, loss function of MRE, Adam optimizer with learning rate of $4 \times 10^{-4}$. To evaluate multi-scale forecasting ability, we perform prediction at 24, 72, 120 autoregressive steps. Performance is measured using anomaly correlation coefficient (ACC) and latitude-weighted mean square error (MSE) (Nguyen et al., 2023) (details in Appendix C.3), reported in Table 2 and 8 (details in Appendix C.3). Besides, we conduct additional experiments at different resolution ($64 \times 128$) and the results are summarized in the Table 6 in

Table 2: ACC↑ (in %) on WeatherBench for six variables and their average at 1, 3, and 5 days. Rows grouped by forecast horizon. **Bold** indicates best performance in each row group. Gray blocks indicate the ablation experiments.

| Method | Operator | Params | Prediction Variables | | | | | | Average |
|---|---|---|---|---|---|---|---|---|---|
| | | | 2T | 10U | 10V | U600 | V600 | T600 | |
| **1 Day Forecast** | | | | | | | | | |
| Climax (Nguyen et al., 2023) | - | 5.4 M | 96.2 ± 2.01 | **91.7 ± 0.67** | **91.4 ± 0.75** | **92.5 ± 0.67** | **90.4 ± 0.82** | **96.2 ± 1.38** | **93.1 ± 0.47** |
| FourCastNet (Pathak et al., 2022) | FNO | 5.3 M | 96.0 ± 2.12 | 90.5 ± 0.75 | 90.0 ± 0.85 | 91.9 ± 0.78 | 90.1 ± 0.95 | 95.5 ± 1.62 | 92.3 ± 0.52 |
| SFNONet (Bonev et al., 2023) | SFNO | 5.3 M | 95.2 ± 2.53 | 81.7 ± 1.81 | 82.1 ± 1.78 | 86.4 ± 1.50 | 84.0 ± 2.02 | 94.1 ± 2.39 | 87.3 ± 0.83 |
| SHNet (Ours) | GSNO | 4.0 M | **96.8 ± 2.00** | 88.3 ± 1.12 | 88.5 ± 1.10 | 90.1 ± 1.07 | 89.3 ± 1.29 | 96.0 ± 1.78 | 91.5 ± 0.59 |
| SFNONet (Bonev et al., 2023) | GSNO | 5.5 M | 95.6 ± 2.37 | 84.3 ± 1.62 | 84.1 ± 1.70 | 87.6 ± 1.35 | 85.8 ± 1.66 | 94.7 ± 2.21 | 88.7 ± 0.76 |
| SHNet | SFNO | 3.7 M | 96.0 ± 2.21 | 86.9 ± 1.47 | 85.8 ± 1.50 | 88.3 ± 1.29 | 86.9 ± 1.42 | 95.3 ± 1.92 | 89.9 ± 0.68 |
| **3 Day Forecast** | | | | | | | | | |
| Climax (Nguyen et al., 2023) | - | 5.4 M | 90.0 ± 5.69 | 58.1 ± 5.08 | 56.3 ± 5.12 | 66.8 ± 4.62 | 57.5 ± 5.30 | 83.4 ± 8.04 | 68.7 ± 2.35 |
| FourCastNet (Pathak et al., 2022) | FNO | 5.3 M | 88.7 ± 6.59 | 56.3 ± 4.83 | 54.4 ± 4.72 | 66.0 ± 4.40 | 55.2 ± 5.40 | 81.8 ± 8.49 | 67.1 ± 2.41 |
| SFNONet (Bonev et al., 2023) | SFNO | 5.3 M | 90.9 ± 5.32 | 58.0 ± 5.39 | 55.2 ± 5.28 | 67.2 ± 4.59 | 57.8 ± 5.66 | 83.4 ± 8.30 | 68.8 ± 2.40 |
| SHNet (Ours) | GSNO | 4.0 M | **92.5 ± 4.63** | **61.7 ± 4.68** | **60.8 ± 4.72** | **70.2 ± 4.21** | **63.6 ± 5.01** | **85.8 ± 7.46** | **72.4 ± 2.13** |
| SFNONet (Bonev et al., 2023) | GSNO | 5.5 M | 91.4 ± 4.97 | 59.8 ± 5.03 | 58.7 ± 4.95 | 68.4 ± 4.42 | 60.1 ± 5.40 | 84.3 ± 7.92 | 70.5 ± 2.27 |
| SHNet | SFNO | 3.7 M | 91.8 ± 4.82 | 60.6 ± 4.90 | 57.2 ± 5.06 | 68.7 ± 4.44 | 61.0 ± 5.22 | 84.6 ± 7.85 | 70.7 ± 2.24 |
| **5 Day Forecast** | | | | | | | | | |
| Climax (Nguyen et al., 2023) | - | 5.4 M | 84.0 ± 9.63 | 28.2 ± 9.35 | 21.3 ± 8.23 | 35.7 ± 10.0 | 14.2 ± 8.57 | 66.3 ± 18.4 | 41.6 ± 4.59 |
| FourCastNet (Pathak et al., 2022) | FNO | 5.3 M | 83.0 ± 11.3 | 29.2 ± 8.68 | 22.5 ± 7.50 | 36.2 ± 10.5 | 13.8 ± 6.47 | 64.5 ± 19.9 | 41.5 ± 4.74 |
| SFNONet (Bonev et al., 2023) | SFNO | 5.3 M | 86.9 ± 7.76 | 35.8 ± 8.74 | 29.5 ± 9.28 | 45.3 ± 9.33 | 26.5 ± 9.47 | 71.2 ± 16.0 | 49.2 ± 4.27 |
| SHNet (Ours) | GSNO | 4.0 M | **88.2 ± 7.21** | **40.3 ± 8.32** | **35.2 ± 8.37** | **47.7 ± 8.71** | **30.7 ± 9.29** | **73.0 ± 14.8** | **52.5 ± 3.99** |
| SFNONet (Bonev et al., 2023) | GSNO | 5.5 M | 87.7 ± 7.68 | 38.6 ± 8.46 | 33.5 ± 8.72 | 47.1 ± 8.92 | 28.9 ± 9.35 | 72.1 ± 15.3 | 51.3 ± 4.11 |
| SHNet | SFNO | 3.7 M | 87.9 ± 7.40 | 38.2 ± 8.62 | 32.3 ± 8.85 | 46.3 ± 9.11 | 28.5 ± 9.36 | 72.4 ± 15.1 | 50.9 ± 4.10 |

Appendix. Our method achieves the competitive results on all variables and time scales, especially in the predictions for the third and fifth days. The wind velocity predictions (V600) and their residual plots (darker plots are better) also demonstrate the accuracy and stability of our method in Figure 4.

To further evaluate superiority of the spectral kernel $G_{\theta_2}^2(l, m)$ as the spectral embedding in GSNO, we construct extra operator-level comparison experiments, with all other architectures and settings identical for fair comparison. The results in the Table 3 clearly demonstrate that our GSNO consistently and significantly outperforms the SFNOs with higher capacity and spatial position-embedding SNO ($G_\theta(x, y)$) across all forecast lead times (details in Appendix C.3).

## 5.3 DIFFUSION MRI MODELLING OF BRAIN MICROSTRUCTURE

Modeling brain microstructure with diffusion MRI is challenging due to sparse, anisotropic measurements across spherical shells (Van Essen et al., 2013; Jeurissen et al., 2014; Zeng et al., 2022); We choose dMRI-based Fiber Orientation Distribution (FOD) angular super-resolution (Zeng et al., 2022; Snoussi & Karimi, 2025) as the specific task for evaluating model performance on anisotropic system; see Appendix C.4 for challenge, motivation and preprocessing details. To further verify GSNO, we adopt the same architecture as FODNet (Zeng et al., 2022), only changing the convolutional layers, adapting it to SFNOs to obtain *FOD-SFNO*, and GSNOs to obtain *FOD-GSNO*. Angular Correlation Coefficient (ACC) is used as an evaluation metric for measuring the angular similarity between two spherical functions (Zeng et al., 2022) (details in Equation 41 in Appendix C.4). The results in Table 4 show that GSNO consistently outperforms other models, demonstrating strong generalization under irregular sampling and data sparsity. Figure 5 shows the region of interest taken from the cingulate gyrus. Compared with the obviously false positive predictions by other methods, the proposed GSNO achieves better prediction of fiber orientation and density.

Table 3: Operator-level comparison on WeatherBench. Bold is the best result.

| Models | Channel | Parameters | Training Time | ACC↑ at 24 h | ACC↑ at 72 h | ACC↑ at 120 h |
|---|---|---|---|---|---|---|
| SFNO | 64 | 3.69 M | 981 s | 89.9% | 70.7% | 50.9% |
| SFNO with 80 channels | 80 | 5.76 M | 1138 s | 89.7% | 70.6% | 51.0% |
| SFNO with 96 channels | 96 | 8.30 M | 1267 s | 89.6% | 70.2% | 50.7% |
| Spatial position-embedding SNO | 64 | 4.31 M | 1037 s | 90.1% | 70.5% | 50.3% |
| **GSNO (ours)** | 64 | 4.00 M | 1024 s | **91.5%** | **72.4%** | **52.5%** |

Figure 5: Predicted dMRI fiber orientation distribution samples of different methods in the same region of cingulate gyrus. Zoom in for clarity and the fibers at the same position have the same visualization perspective for clear fiber orientation and density comparison.

Table 4: Performance comparison on HCP dataset (ACC↑). Bold represents the best result.

| Methods | SSMT-CSD (Khan et al., 2020) | FOD-Net (Zeng et al., 2022) | FOD-SFNO (Bonev et al., 2023) | ESCNN (Snoussi & Karimi, 2025) | **FOD-GSNO (Ours)** |
|---|---|---|---|---|---|
| Parameters | – | 19.44 M | 1.15 M | 1.47 M | 1.21 M |
| White matter | $0.7523 \pm 0.0256$ | $0.8858 \pm 0.0138$ | $0.8995 \pm 0.0151$ | $0.9006 \pm 0.0142$ | $\mathbf{0.9083 \pm 0.0140}$ |
| Whole brain | $0.6640 \pm 0.0145$ | $0.8250 \pm 0.0159$ | $0.8334 \pm 0.0154$ | $0.8362 \pm 0.0162$ | $\mathbf{0.8517 \pm 0.0139}$ |

## 5.4 Ablation Experiments and computational costs

To further evaluate the contributions of GSNO and SHNet, we conduct ablation studies to replace the GSNO block with a classic SFNO block in the SHNet or replace the classic SFNO block with GSNO block in the SFNONet, leading to performance degradation across all variables and time horizons (Table 2 and 8 in Appendix C.3). This confirms the effectiveness of GSNO and the multi-scale architecture of SHNet. Further, we observe more pronounced performance gains from GSNO blocks in longer-term predictions, suggesting that GSNO plays an increasingly important role in modelling long-range dynamics. To demonstrate the efficiency comparison, we provide the parameters of different models in Table 2. Detailed computational costs of different models are in Appendix C.6.

We provide extra ablation experiments and interpretability analysis of the original term and correction term in our proposed GSNO on weather prediction (details in Appendix C.5). As shown in the Figure 6 and Table 11, even if $I(\mathcal{T}_{\mathrm{orig}})$ term is frozen, using only $I(\mathcal{T}_{\mathrm{corr}})$ term clearly outlines the temperature distribution framework linked to the Earth's topography, distinguishing the persistent low temperatures at the poles from the high-temperature pattern in regions like the equator. This explicitly demonstrates the effective representation of the $I(\mathcal{T}_{\mathrm{corr}})$ for the topography constraints.

## 6 Conclusion

We have introduced a rigorous theoretical framework for spherical neural operators, grounded in the formulation of **designable Green's function** on the sphere, and extended to the spherical harmonic domain. This framework is used to design different spherical operators and enables the principled incorporation of complex system constraint on spherical operator learning. Building on this framework, we design an absolute and relative position-dependent Green's function, yielding a novel spherical operator solution beyond existing equivariant operator solution to capture complex physical constraints for asymmetric conditions and position-dependent phenomena. The proposed Green's function leads to GSNO, designed to handle a broader range of real-world scenarios by the flexible balance between equivariance and invariance while retaining spherical geometry and grid invariance. Further, we design a multi-scale architecture, SHNet, incorporating geometrically adaptive up-and-down sampling to enhance interactions across resolutions on the sphere. Both theoretical analysis and extensive experiments on multiple spherical scenarios consistently demonstrate the superiority of our approaches over many state-of-the-art methods. This work establishes a unified, system-level perspective for designing spherical operator through designable Green's functions, suggesting further designs for the development of spherical modeling.

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

# A    SPHERICAL CONVOLUTION THEOREM

## A.1    PROPERTIES OF SPHERICAL HARMONICS

The explicit form of spherical harmonic function (Müller, 2006) is:

$$Y_l^m(\theta, \phi) = \sqrt{\frac{(2l+1)(l-m)!}{4\pi(l+m)!}} P_l^m(\cos\theta) e^{im\phi} \tag{21}$$

Spherical harmonic functions satisfy the orthonormality condition (Müller, 2006):

$$\int_{S^2} Y_l^m(\omega) \overline{Y_{l'}^{m'}(\omega)} \, d\omega = \delta_{ll'} \delta_{mm'}. \tag{22}$$

where $\delta$ is the Kronecker delta.

For subsequent analysis, we recall how the spherical harmonics transform under a rotation operation $R$ from $SO(3)$. The rotation of the spherical harmonic function can be expressed as (Driscoll & Healy, 1994):

$$Y_l^m(R\omega) = \Lambda(R^{-1})Y_l^m(\omega) = \sum_{|k|\leq l} D_{k,m}^{(l)}(R^{-1})Y_l^k(\omega), \tag{23}$$

where $D_{k,m}^{(l)}$ denotes the Wigner D-matrix (Edmonds, 1996) and $\Lambda(R)$ represents the rotation operator acting on spherical functions. Taking the complex conjugate of Equation equation 23 yields:

$$\overline{Y_l^m(R\omega)} = \sum_{|k|\leq l} \overline{D_{k,m}^{(l)}(R^{-1})} \, \overline{Y_l^k(\omega)}. \tag{24}$$

## A.2    INTERCHANGE OF INTEGRALS AND VARIABLE SUBSTITUTION

Interchange integral order after substituting the convolution definition into the spherical harmonic transform:

$$\mathrm{SHT}[(f*h)](l,m) = \int_{S^2} \left( \int_{SO(3)} f(Rn)h(R^{-1}\omega) \, dR \right) \overline{Y_l^m(\omega)} \, d\omega$$

$$= \int_{SO(3)} f(Rn) \left( \int_{S^2} h(R^{-1}\omega) \overline{Y_l^m(\omega)} \, d\omega \right) dR. \tag{25}$$

Let $\omega' = R^{-1}\omega$, then $\omega = R\omega'$ with $d\omega = d\omega'$ (measure preservation under rotation). The inner integral becomes:

$$\int_{S^2} h(\omega') \overline{Y_l^m(R\omega')} \, d\omega'. \tag{26}$$

Substituting the expansion from Equation equation 24:

$$\int_{S^2} h(\omega') \sum_{|k|\leq l} \overline{D_{k,m}^{(l)}(R^{-1})} \, \overline{Y_l^k(\omega')} \, d\omega' = \sum_{|k|\leq l} \overline{D_{k,m}^{(l)}(R^{-1})} \cdot \mathrm{SHT}[h](l,k). \tag{27}$$

## A.3    ORTHOGONALITY CONDITION SCREENING NON-ZERO TERMS

Substituting Equation equation 27 into Equation equation 25 gives:

$$\mathrm{SHT}[(f*h)](l,m) = \sum_{|k|\leq l} \mathrm{SHT}[h](l,k) \int_{SO(3)} f(Rn) \overline{D_{k,m}^{(l)}(R^{-1})} \, dR. \tag{28}$$

Parameterize $R \in SO(3)$ using Euler angles $R = u(\phi)a(\theta)u(\psi)$ (Driscoll & Healy, 1994), with inverse $R^{-1} = u(-\psi)a(-\theta)u(-\phi)$. The corresponding D-matrix is:

$$D_{k,m}^{(l)}(R^{-1}) = e^{ik\phi} d_{k,m}^{(l)}(cos(\theta)) e^{im\psi}. \tag{29}$$

which d is the Wigner d-matrix (Edmonds, 1996). Then Taking the complex conjugate:

$$\overline{D_{k,m}^{(l)}(R^{-1})} = e^{-ik\phi} d_{k,m}^{(l)}(cos(\theta)) e^{-im\psi}. \tag{30}$$

For any rotation $u(\psi)$ on the z-axis, substitute $R \to Ru(\psi)$. Due to right-invariance of the Haar measure $dR$ (Driscoll & Healy, 1994), the remaining integral is adjusted to:

$$
\begin{aligned}
I &= \int_{SO(3)} f(Rn) \overline{D_{k,m}^{(l)}(R^{-1})} \, dR \\
&= \int_{SO(3)} f(Ru(\psi)n) \, \overline{D_{k,m}^{(l)}\left((Ru(\psi))^{-1}\right)} \, dR \\
&= \int_{SO(3)} f(Ru(\psi)n) \, \overline{D_{k,m}^{(l)}\left(u(-\psi)R^{-1}\right)} \, dR \\
&= \int_{SO(3)} f(Ru(\psi)n) \, e^{-ik\psi} \overline{D_{k,m}^{(l)}(R^{-1})} \, dR \\
&= e^{-ik\psi} \int_{SO(3)} f(Rn) \overline{D_{k,m}^{(l)}(R^{-1})} \, dR \\
&= e^{-ik\psi} I
\end{aligned} \tag{31}
$$

From the above equation:

$$I \left(1 - e^{-ik\psi}\right) = 0.$$

Therefore, if $k \neq 0$, choose $\psi$ such that $1 - e^{-ik\psi} \neq 0$, forcing $I = 0$. The only nontrivial solution is $j = 0$, where $e^{-ik\psi} = 1$.

Therefore, **only the $k = 0$ term survives.** So the overall integral is simplified to:

$$\text{SHT}[(f * h)](l, m) = \text{SHT}[h](l, 0) \int_{SO(3)} f(Rn) \overline{D_{0,m}^{(l)}(R^{-1})} \, dR. \tag{32}$$

### A.4 RELATIONSHIP BETWEEN D-MATRICES AND SPHERICAL HARMONICS

Because the unitary characteristic of the Wigner D-matrix (Driscoll & Healy, 1994), we can get the relationship between D-matrix and spherical Harmonic function:

$$\overline{D_{0,m}^{(l)}(R^{-1})} = D_{m,0}^{(l)}(R) = \sqrt{\frac{4\pi}{2l+1}} \overline{Y_l^m(\theta, \phi)} \tag{33}$$

### A.5 COMPLETE PROOF OF SPHERICAL CONVOLUTION THEOREM

Based on all the above theories, the complete derivation process is as follows:

$$
\begin{aligned}
\text{SHT}[(f * h)](l, m) &= \int_{S^2} \left( \int_{SO(3)} f(Rn) h(R^{-1}\omega) \, dR \right) \overline{Y_l^m(\omega)} \, d\omega \\
&= \int_{SO(3)} f(Rn) \left( \int_{S^2} h(\omega) \overline{Y_l^m(R\omega)} \, d\omega \right) dR \\
&= \text{SHT}[h](l, k) \int_{SO(3)} f(Rn) \overline{D_{k,m}^{(l)}(R^{-1})} \, dR \\
&= \text{SHT}[h](l, 0) \int_{SO(3)} f(Rn) \overline{D_{0,m}^{(l)}(R^{-1})} \, dR \\
&= \text{SHT}[h](l, 0) \int_{SO(3)} f(Rn) \sqrt{\frac{4\pi}{2l+1}} \overline{Y_l^m(\theta, \phi)} \, dR \\
&= 2\pi \sqrt{\frac{4\pi}{2l+1}} \cdot \text{SHT}[h](l, 0) \cdot \text{SHT}[f](l, m).
\end{aligned} \tag{34}
$$

# B  DETAILS OF OUR MODEL

## B.1  COMPLETE OPERATOR DERIVATION VIA SPHERICAL HARMONIC EXTENSION

The spherical harmonic form of Green's function is:

$$G(u) = \sum_{l=0}^{\infty} \sum_{m=-l}^{l} \text{SHT}[G](l,m)Y_l^m(u). \tag{35}$$

The detailed derivation process is as follows:

$$
\begin{aligned}
\text{SHT}[g(u)](l,m) &= \int_{SO(3)} f(Rn) \left( \int_{S^2} G(R^{-1}u)\overline{Y_l^m(u)}\, du \right) dR \\
&= \int_{SO(3)} f(Rn) \left( \int_{S^2} \sum_{|m'| \le l'} \text{SHT}[G](l',m')Y_{l'}^{m'}(R^{-1}u)\overline{Y_l^m(u)}\, du \right) dR \\
&= \text{SHT}[G](l',m') \int_{SO(3)} f(Rn) \left( \int_{S^2} Y_{l'}^{m'}(u)\overline{Y_l^m(Ru)}\, du \right) dR \\
&= \text{SHT}[G](l',m') \int_{SO(3)} f(Rn) \left( \int_{S^2} Y_{l'}^{m'}(u)) \sum_{|k| \le l} \overline{D_{k,m}^{(l)}(R^{-1})}\,\overline{Y_l^k(u)}\, du \right) dR \\
&= \text{SHT}[G](l',m')\delta_{l'l}\delta_{m'k} \int_{SO(3)} f(Rn)\overline{D_{k,m}^{(l)}(R^{-1})}\, dR \\
&= \text{SHT}[G](l,0) \int_{SO(3)} f(Rn)\overline{D_{0,m}^{(l)}(R^{-1})}\, dR \\
&= 2\pi\sqrt{\frac{4\pi}{2l+1}} \cdot \text{SHT}[G](l,0) \cdot \text{SHT}[f](l,m). \tag{36}
\end{aligned}
$$

## B.2  SPHERICAL HARMONIC NEURAL OPERATOR NETWORK

For WB and SSWE experiments, this study uses Spherical Harmonic Neural Operator Network (SHNet) with a depth of 2, which performs two spherical downsampling operations. Specifically, the encoder first expands the input channel to $C$. In the first two GSNO blocks, the spatial resolution, i.e., the number of latitudinal and longitudinal sampling points in the inverse spherical harmonic transform (ISHT), is reduced by half, while the channel dimension is doubled via MLPs and convolutional layers.

Subsequently, the next two GSNO blocks perform upsampling by doubling the number of samples in latitude and longitude while continuing to transform the channels. The final GSNO block is used as the output layer: it projects the features back to channel dimension $C$ without applying further spatial scaling. The implementations of MLPs, encoders and decoders are consistent with those used in SFNONet (Bonev et al., 2023). The kernel of all convolution operations is $1 \times 1$ for channel adjustment. SHNet also includes three skip connections to enhance gradient flow and multi-scale feature fusion: The output of the final GSNO block is concatenated with the original input; The output of the fourth GSNO block is added to the input of the first GSNO block; The output of the third GSNO block is added to the input of the second GSNO block.

# C  EXPERIMENTAL DETAILS

## C.1  MODEL IMPLEMENTATION DETAIL

The key model parameter settings used for the two experiments are presented in Table 5. Our proposed SHNet with a depth of 2 consists of five blocks, including two downsampling layers, two upsampling layers, and an output layer. Furthermore, the embedding dimension of SHNet is set to a smaller value (8 and 64), as the depth increases, the embedding dimension gradually increases until

it is four times the original dimension. The depth of other models represents the number of core blocks.

Table 5: Key model hyper-parameters on Spherical Shallow Water Equations (SSWE) and Weather-Bench (WB) experiments.

| Hyperparameters | ClimaX | | FourCastNet | | SFNONet | | SHNet | |
|---|---|---|---|---|---|---|---|---|
| | SSWE | WB | SSWE | WB | SSWE | WB | SSWE | WB |
| Depth | 6 | 6 | 8 | 8 | 4 | 4 | 2 | 2 |
| Embedding dimension | 128 | 256 | 128 | 384 | 32 | 256 | 8 | 64 |
| Activation function | GELU | GELU | GELU | GELU | GELU | GELU | GELU | GELU |
| MLP ratio | 2 | 2 | 2 | 2 | 2 | 2 | 2 | 2 |
| Patch size | $4 \times 4$ | $4 \times 4$ | $4 \times 4$ | $4 \times 4$ | / | / | / | / |
| Parameters | 1.01 M | 5.4 M | 1.37 M | 5.3 M | 0.28 M | 5.3 M | 0.80 M | 4.0 M |

The loss function $\mathcal{L}$ of the proposed method is the spherical grid weighted mean relative error between the prediction $F(X_n)$ and the target $Y_{n+t}$, calculated following Bonus *et al.* (Bonev et al., 2023) as follows:

$$\mathcal{L}[F(X_n), Y_{n+t}] = \frac{1}{C} \sum_{c=1}^{C} \left( \frac{\sum_{i,j} v_{i,j} |F(X_n)(x_{c,i,j}) - Y_{n+t}(x_{c,i,j})|^2}{\sum_{i,j} v_{i,j} |Y_{n+t}(x_{c,i,j})|^2} \right)^{\frac{1}{2}}, \tag{37}$$

where $v_{i,j}$ is the products of the Jacobian $sin(\lambda_i)$ ($\lambda_i$ represents the latitude at grid point) and the quadrature weights (Bonev et al., 2023). $C$ is the number of predicted variables. $n$ is the index of the initial time step, $t$ is the predicted autoregressive steps. This loss function is used in all the experiments.

## C.2 SPHERICAL SHALLOW WATER EQUATIONS

Spherical Shallow Water Equations (SSWE) form a nonlinear hyperbolic PDE system that models the motion of thin-layer fluids on a rotating sphere. The core underlying assumption is the shallow water approximation, where the vertical scale of the fluid layer is much smaller than the horizontal scale (Bonev et al., 2018).

The dataset is publicly available, which can be obtained and used for evaluation through Torch-Harmonics GitHub (Bonev et al., 2023). Consistent with Bonus *et al.* (Bonev et al., 2023), we use mean relative error of each variable as the evaluation metric.

We also conduct the extra experiments on SSWE under the exact same setting as that of SFNO for reference (e.g., 150 time steps). The results are in Table 7.

## C.3 WEATHER FORECASTING

The dataset is publicly available at WeatherBench GitHub (Rasp et al., 2020). Following (Liu et al., 2024), the chosen 24 common variables are detailed in Table 9, including 10U, 10V, 2T, U50, U50, U250, U500, U600, U700, U850, U925, V50, V250, V500, V600, V700, V850, V925, T50, T250, T500, T600, T700, T850, T925. And 10U, 10V, 2T, U600, V600 and T600 are prediction target.

And the additional results (MSE) are in Table 8.

The anomaly correlation coefficient (ACC) and the latitude-weighted mean square error (MSE) are used to measure the performance of the model in WeatherBench (WB). MSE between the prediction $F(X_n)$ and the target $Y_{n+t}$ for evaluation is calculated following Rasp *et al.* (Nguyen et al., 2023) as:

$$\text{MSE}[F(X_n), Y_{n+t}] = \frac{1}{C \times H \times W} \sum_{c=1}^{C} \sum_{i=1}^{H} \sum_{j=1}^{W} w_i \left( F(X_n)(x_{c,i,j}) - Y_{n+t}(x_{c,i,j}) \right)^2 \tag{38}$$

Table 6: Supplementary experiments on SSWE at $128 \times 256$ resolution (MRE↓) and WB at $64 \times 128$ resolution (ACC↑).

| Methods | SSWE at 5h | SSWE at 10h | WB at 1day | WB at 3day | WB at 5day |
|---------|-----------|-------------|------------|------------|------------|
| FNO | 2.22 | 2.73 | 93.4 | 71.5 | 42.5 |
| SFNO | 0.74 | 0.87 | 91.6 | 73.2 | 49.2 |
| GSNO | **0.68** | **0.79** | **93.0** | **74.9** | **51.7** |

Table 7: Experimental results ($L^2$) on SSWE under the exact same setting for reference (e.g., 150 time steps).

| Models | FNO | SFNO | GSNO |
|--------|-----|------|------|
| At 1 h | $8.628 \times 10^{-4}$ | $8.092 \times 10^{-4}$ | $7.051 \times 10^{-4}$ |
| At 10 h | $9.470 \times 10^{-3}$ | $6.739 \times 10^{-3}$ | $5.178 \times 10^{-3}$ |

where $H$ is the number of latitudes and $W$ is the number of longitudes. The latitude weight (Nguyen et al., 2023) are calculated as:

$$w_i = \frac{\cos(\lambda_i)}{\frac{1}{H} \sum_{i=1}^{H} \cos(\lambda_i)} \tag{39}$$

ACC (Nguyen et al., 2023) is used to measure spatial correlation between predicted anomalies $F(X_n)'$ and target anomalies $Y'_{n+t}$:

$$\text{ACC} = \frac{\sum_{c,i,j} w_i \left( F(X_n)'_{c,i,j} \cdot (Y_{n+t})'_{c,i,j} \right)}{\sqrt{\sum_{c,i,j} w_i \left( F(X_n)'_{c,i,j} \right)^2 \cdot \sum_{c,i,j} w_i \left( (Y_{n+t})'_{c,i,j} \right)^2}} \tag{40}$$

Besides, to empirically compare the correction term of GSNO with positional embedding, we construct a "Spatial position-embedding SNO" model that strictly implements the $SHT[f + C_f \cdot [G]_{\theta(x,y)}]$ path, maintaining all other architecture and settings identical for a fair comparison.

The results in the Table 3 clearly demonstrate that our spectral implementation (GSNO) consistently and significantly outperforms the SFNO and spatial position-embedding SNO across all forecast lead times. This indicates that the spectral implementation derived from our Green's function framework is, in itself, a more effective and different design. Moreover, the spatial position embedding added in each operator ($SHT[f + C_f \cdot [G]_{\theta(x,y)}]$) is not conducive to long-term stable prediction.

### C.4 DIFFUSION MRI MODELING OF BRAIN MICROSTRUCTURE

**Detailed description**. Diffusion MRI-based FOD angular super resolution in this study is a distinct challenge in: (1) the input signals are sparse and anisotropic diffusion measurements acquired over spherical shells, exhibiting sharp angular variations corresponding to underlying fiber tract orientations; (2) the spatial sampling is performed using HEALPix, resulting in nonuniform and incomplete coverage of the spherical domain.

**The task motivation** of dMRI-FOD rather than dMRI itself: The raw dMRI signals themselves, due to noise, partial volume effects, and the aliasing of signals from multiple tissues, do not directly exhibit clear, anisotropic fiber structures. Learning directly from raw dMRI would require the model to simultaneously handle noise suppression, signal unmixing, and orientation estimation, introducing numerous confounding factors. This would make it difficult to cleanly evaluate the operator's core capability in modeling anisotropic geometric structures. Further, high angular resolution dMRI is costly and often infeasible in clinical settings, leading to poor-quality FODs. Therefore, enhancing dMRI quality from routine dMRI while while handling noise suppressiona and signal unmixing for estimating FOD is a widely recognised and practical problem.

Table 8: MSE↓ (×10⁻³) on WeatherBench for six variables and their average at 1, 3, and 5 days. Rows grouped by forecast horizon. **Bold** indicates best performance within each group. Gray blocks indicate the ablation experiments.

| Method | Operator | Prediction Variables | | | | | | Average |
|---|---|---|---|---|---|---|---|---|
| | | 2T | 10U | 10V | U600 | V600 | T600 | |
| **1 Day Forecast** | | | | | | | | |
| Climax (Nguyen et al., 2023) | - | 4.23 ± 0.43 | **80.9 ± 5.16** | **113 ± 7.04** | **67.9 ± 4.17** | **118 ± 7.71** | **6.68 ± 0.51** | **66.8 ± 2.06** |
| FourCastNet (Pathak et al., 2022) | - | 4.36 ± 0.45 | 94.2 ± 5.95 | 132 ± 8.18 | 78.9 ± 5.17 | 137 ± 9.30 | 7.98 ± 0.66 | 75.7 ± 2.45 |
| SFNONet (Bonev et al., 2023) | SFNO | 5.18 ± 0.58 | 175 ± 12.1 | 242 ± 17.0 | 134 ± 8.98 | 234 ± 16.7 | 10.8 ± 0.93 | 134 ± 4.70 |
| SHNet (Ours) | GSNO | **4.22 ± 0.40** | 124 ± 8.72 | 173 ± 10.4 | 106 ± 7.18 | 168 ± 11.3 | 8.75 ± 0.81 | 97.5 ± 3.18 |
| SFNONet (Bonev et al., 2023) | GSNO | 4.82 ± 0.52 | 162 ± 10.5 | 221 ± 14.9 | 127 ± 8.32 | 215 ± 14.8 | 9.97 ± 0.87 | 123 ± 4.16 |
| SHNet | SFNO | 4.76 ± 0.48 | 156 ± 9.90 | 208 ± 14.1 | 122 ± 8.34 | 204 ± 14.1 | 9.89 ± 0.86 | 117 ± 3.97 |
| **3 Day Forecast** | | | | | | | | |
| Climax (Nguyen et al., 2023) | - | 11.6 ± 1.55 | 384 ± 35.0 | 545 ± 56.1 | 305 ± 28.8 | 558 ± 61.4 | 31.4 ± 4.38 | 306 ± 15.8 |
| FourCastNet (Pathak et al., 2022) | - | 12.7 ± 1.66 | 402 ± 33.1 | 594 ± 54.4 | 322 ± 28.0 | 616 ± 62.3 | 33.9 ± 4.51 | 330 ± 15.6 |
| SFNONet (Bonev et al., 2023) | SFNO | 10.3 ± 1.51 | 372 ± 32.0 | 535 ± 51.0 | 295 ± 26.7 | 546 ± 60.5 | 30.3 ± 4.34 | 298 ± 14.9 |
| SHNet (Ours) | GSNO | **9.08 ± 1.32** | **335 ± 26.9** | **478 ± 44.7** | **267 ± 24.0** | **492 ± 51.4** | **26.7 ± 3.79** | **268 ± 13.9** |
| SFNONet (Bonev et al., 2023) | GSNO | 9.59 ± 1.45 | 357 ± 29.3 | 501 ± 48.1 | 281 ± 25.5 | 524 ± 56.2 | 28.5 ± 4.10 | 284 ± 14.0 |
| SHNet | SFNO | 9.50 ± 1.40 | 362 ± 29.3 | 495 ± 47.7 | 284 ± 25.9 | 522 ± 55.8 | 28.1 ± 3.91 | 282 ± 13.9 |
| **5 Day Forecast** | | | | | | | | |
| Climax (Nguyen et al., 2023) | - | 18.0 ± 3.15 | 572 ± 53.6 | 831 ± 91.2 | 518 ± 49.4 | 922 ± 121 | 57.7 ± 9.46 | 486 ± 28.1 |
| FourCastNet (Pathak et al., 2022) | - | 20.9 ± 3.18 | 510 ± 41.7 | 725 ± 70.7 | 471 ± 39.3 | 811 ± 99.5 | 58.9 ± 8.24 | 433 ± 22.5 |
| SFNONet (Bonev et al., 2023) | SFNO | 14.4 ± 2.57 | 502 ± 46.5 | 726 ± 81.6 | 440 ± 42.9 | 794 ± 105 | 48.1 ± 7.76 | 421 ± 24.6 |
| SHNet (Ours) | GSNO | **13.3 ± 2.34** | **471 ± 42.2** | **687 ± 73.4** | **411 ± 40.5** | **762 ± 93.8** | **45.9 ± 7.32** | **398 ± 22.2** |
| SFNONet (Bonev et al., 2023) | GSNO | 13.9 ± 2.46 | 484 ± 44.0 | 699 ± 75.1 | 427 ± 41.4 | 776 ± 94.9 | 47.1 ± 7.57 | 408 ± 22.6 |
| SHNet | SFNO | 13.8 ± 2.45 | 492 ± 44.8 | 705 ± 76.3 | 430 ± 41.5 | 783 ± 96.0 | 46.8 ± 7.54 | 412 ± 22.9 |

Table 9: Chosen Variables Abbreviation

| Abbreviation | Description |
|---|---|
| 10U | Zonal wind velocity at 10m from the surface |
| 10V | Meridional wind velocity at 10m from the surface |
| 2T | Temperature at 2m from the surface |
| U−− | Zonal wind velocity at pressure level −− |
| V−− | Meridional wind velocity at pressure level −− |
| T−− | Temperature at pressure level −− |

We randomly select dMRI images of 30 subjects in the Human Connectome Project (HCP) (Van Essen et al., 2013), where 20 subjects were for training, 5 for validation, and 5 for test. **Detailed pre-processing**. The complete dMRI data of each subject contains 288 volumes (multi-shell HARDI), including 270 volumes of b=1000, 2000, 3000 $s/mm^2$ with 90 gradient directions for each shell and 18 b0 volumes. Due to the wide application of low b-value with 32 gradient directions in clinical practice (Zeng et al., 2022), we subsample 32 volumes of b=1000 and 1 volume of b0 from the complete dMRI according to the HCP protocol, to obtain single-shell LARDI. Further, we use MSMT-CSD (Jeurissen et al., 2014) on the multi-shell HARDI to obtain high angular resolution fiber orientation distribution (FOD) as the ground truth (HAR-FOD). Meanwhile, we use SSMT-CSD (Khan et al., 2020) on the single-shell LARDI to obtain single-shell low angular resolution FOD as the condition of GSNO (LAR-FOD). $l_{max}$ is set to the default value of 8 to balance precision and complexity (Zeng et al., 2022).

SSMT-CSD (Khan et al., 2020), FOD-Net (Zeng et al., 2022), FOD-SFNO (Bonev et al., 2023), ESCNN (Snoussi & Karimi, 2025) and our proposed FOD-GSNOadopt the same network architecture as FOD-Net. The difference lies in that FOD-Net uses 3D convolutional layers to handle a large number of voxels, while FOD-SFNO, ESCNN and FOD-GSNO stack voxels on the channel dimension and then use the corresponding operators for processing. To ensure a fair comparison, FOD-SFNO and FOD-GSNO have exactly the same hyperparameters, with the only difference being the types of operators. And their parameter comparison is shown in Table 10. Angular Correlation

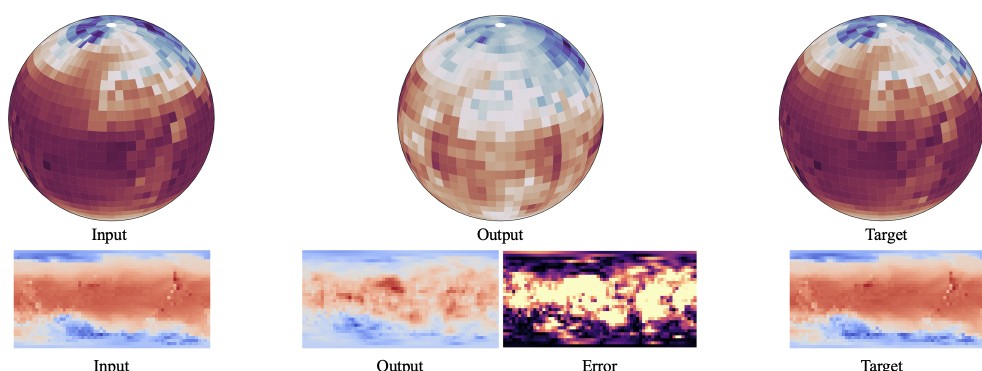

Figure 6: Predicted temperature field at 2m from the earth surface of GSNO w/o I($\mathcal{T}_{\text{orig}}$). Provide the residual (error) to the ground truth for clarity (darker is better).

Coefficient (ACC) is used for evaluation:

$$\mathbf{ACC}(u, \nu) = \frac{\sum_{l=0}^{8} \sum_{m=-l}^{l} g'_{lm} g^*_{lm}}{\left(\sum_{l=0}^{8} \sum_{m=-l}^{l} g^2_{lm}\right)^{\frac{1}{2}} \left(\sum_{l=0}^{8} \sum_{m=-l}^{l} g^2_{lm}\right)^{\frac{1}{2}}} \tag{41}$$

where $l, m$ represent a degree and order of a spherical harmonic function. $g', g$ are the ground-truth (MSMT-CSD) and the estimated FOD from different methods.

Table 10: Parameters on dMRI.

| Models | FOD-Net | FOD-SFNO | ESCNN | FOD-GSNO |
|---|---|---|---|---|
| Parameters | 19.44 M | 1.15 M | 1.47 M | 1.21 M |

## C.5 EXTRA EXPERIMENTS AND INTERPRETABILITY ANALYSIS

The additive form in Equation 20 allows us to explicitly decouple the rotationally equivariant component. This separation reflects real-world scenarios where the global physical dynamics (e.g., geophysical flow) are predominantly symmetric, but local features (e.g., terrain, boundary anomalies) introduce perturbative effects. The additive form lets us modulate these independently and retain interpretability. Overall, this allows a clean decomposition and analysis of each term, as validated in Tables 11. As shown in Figure 6, even if I($\mathcal{T}_{\text{orig}}$) term is frozen, using only I($\mathcal{T}_{\text{corr}}$) term clearly outlines the fundamental temperature distribution framework linked to the Earth's topography, successfully distinguishing the persistent low temperatures at the poles from the overall high-temperature pattern in regions like the equator. This explicitly demonstrates the effective representation of the correction term for the topography constraints. Furthermore, the finer undulations in high-temperature regions reflect stable, large-scale, non-equivariant patterns present in the underlying temperature field and in its discretization, including: (1) Climatological zonal structure: temperature fields exhibit persistent, slowly varying zonal (east–west) asymmetries arising from longitudinal land–ocean contrasts and stationary planetary waves. (2) Latitude-dependent variability: numerical discretization on an equiangular grid leads to resolution and dissipation patterns that vary with latitude. (3) Long-term statistical structure: even though instantaneous dynamics vary, the climatological mean temperature field (which the model implicitly learns) contains smooth meridional and zonal gradients.

## C.6 DETAILED COMPUTATIONAL TRADE-OFFS

We add the comparison of parameters, FLOPS, training time (seconds per epoch) and inference time for two models, SFNONet and SHNet. All results are obtained using the same hardware and dataset (Weatherbench).

Table 11: The ablation experiments of $I(\mathcal{T}_{orig})$ and $I(\mathcal{T}_{corr})$ items in GSNO are conducted to verify the effectiveness of $I(\mathcal{T}_{orig})$. The average ACC (%) of various variables is presented.

| Models | GSNO | GSNO w/o $I(\mathcal{T}_{corr})$ | GSNO w/o $I(\mathcal{T}_{orig})$ |
|--------|------|------|------|
| day 1 | 91.5 | 71.4 | 20.1 |
| day 3 | 72.4 | 61.3 | 11.1 |
| day 5 | 52.5 | 42.6 | 9.9 |

In the second and third columns, we compare the operators. GSNO introduces a lightweight set of additional global spherical integration term compared to the standard SFNO. This results in a modest increase in model size and computational cost. Specifically, GSNO adds approximately 1.6% to the runtime, along with a slight increase in parameters and FLOPs, while delivering around a 6% improvement in performance. In the comparison between the second and fourth columns, where the same operator is used but different networks are employed, SHNet demonstrates fewer parameters and lower FLOPs than SFNONet, owing to its multi-scale design. The slightly increased runtime in SHNet is primarily due to the added complexity of the GSNO operator, particularly the correction term. Since GSNO is applied at every layer (as shown in Figure 1), this results in an approximate 10% increase in runtime. Nevertheless, this design achieves a 2% performance gain while reducing overall memory and computational usage.

Table 12: Parameters and Inference Time on WeatherBench.

| Models | SHNet (SFNO) | SHNet (GSNO) | SFNONet (SFNO) | SFNONet (GSNO) |
|--------|------|------|------|------|
| Parameters | 3.69 M | 4.00 M | 5.26 M | 5.52 M |
| Training Time | 981 s | 1024 s | 937 s | 983 s |
| Inference Time | 245.66 ms | 249.82 ms | 217.32 ms | 227.18 ms |
| Performance↑ | 50.9 | **52.5** | 49.2 | 51.3 |

Table 13: Extra performance and efficiency comparison, SFNO and GSNO use the same U-Net architecture for fair comparison.

| Models | SFNO | GSNO | SFNO (80 C) | SFNO (96 C) | FourCastNet (FNO) | Climax |
|--------|------|------|------|------|------|------|
| Channel | 64 | 64 | 80 | 96 | 384 | 256 |
| Parameters | 3.69 M | 4.00 M | 5.76 M | 8.30 M | 5.30 M | 5.40 M |
| Training Time | 981 s | 1024 s | 1138 s | 1267 s | 878 s | 891 s |
| Performance↑ | 50.9 | **52.5** | 51.0 | 50.7 | 41.5 | 41.6 |

## D LIMITATIONS

Our experiments are conducted on the SSWE dataset ($256 \times 256$ resolution), the WB dataset ($5.625°$ resolution) and dMRI modeling of brain microstructure. While these results demonstrate the potential of our method, they are limited in scope, such as other manifolds. The generalizability to other spatial resolutions and datasets remains to be verified. In future work, we plan to evaluate our approach across a wider range of resolutions and on additional spherical scenarios.

## E REPRODUCIBILITY

We include the code for our model in the supplementary material, containing a README.md for guidance. We will release the complete code at `https://github.com/haot2025/GSNO`.

## F THE USE OF LARGE LANGUAGE MODELS (LLMS)

We used Large Language Models (LLMs), such as GPT-4-based systems, as writing polishing assistants during the paper polishing stage to enhance clarity, readability, and precision of our paper. And main contributions, such as scientific content, theoretic analysis, technical contributions, experimental design, and all results of this paper were conceived, executed, and verified by the authors.

