# OpenReview forum: "Generalized Spherical Neural Operators: Green’s Function Formulation"
_ICLR.cc/2026/Conference — ICLR 2026 Poster_

### Official Review · Reviewer_vG13 · 2025-10-20

**Soundness:** 3
**Presentation:** 2
**Contribution:** 2
**Rating:** 4
**Confidence:** 4

**Summary:**

Spherical Fourier Neural Operators (SFNOs) are a form of neural network that take as input a scalar signal on the sphere and output a new scalar signal on the sphere. A major part of SFNOs are spherical convolutions that are equivariant under rotations. This convolution is implemented in the Fourier space of the sphere.

The main idea of the submitted paper is to add positional dependency to the spherical convolution, thus breaking the rotation symmetry.This is done in each convolution layer by adding a learned positional embedding to the signal before it is convolved by a filter. The addition of the positional embedding is also done in Fourier space.

Experiments on three tasks (without full rotational symmetry) show that the added positional information can be beneficial.

**Strengths:**

1. The proposed method is well-motivated, easy to implement and seems to work.
2. The paper is clear and compares extensively to the most relevant baseline, i.e. networks built from ordinary SFNOs.

**Weaknesses:**

I have a few technical questions, see below. Other than that, the main weakness is that relevant prior literature is not sufficiently covered in my opinion.

1. The idea to add auxiliary positional information is a spherical analogue of absolute positional embeddings as used in NLP/vision, and very similar to for instance “CoordConv” (Liu et al, An intriguing failing of convolutional neural networks and the coordconv solution, NeurIPS 2018).
2. More generally, symmetry breaking by adding auxiliary inputs is a broad research direction that is not referenced in the submitted paper. See e.g. Smidt et al (Finding Symmetry Breaking Order Parameters with Euclidean Neural Networks, Phys. Rev. Research 3, 012002 (2021)) and later follow-ups.
3. Spherical CNNs are not mentioned in the related work at all, except a reference to Cohen et al 2018, without explaining that it covers spherical networks. I believe spherical CNNs should be discussed in a bit more detail with references also to later works. When using spherical convolutions such as the ones in the submitted paper, the filters must be zonal (e.g. seen by formula 13, where $m\neq 0$ parts of $G$ are not present). This is addressed in prior work on spherical CNNs for instance by lifting the signal from the sphere to SO(3) or by storing steerable features relative to a local gauge at each point on the sphere (Cohen et al, Gauge Equivariant Convolutional Networks and the Icosahedral CNN, ICML 2019). The potential for obtaining more expressive layers in this way should at least be mentioned in the submitted work.

It would make the contribution of the paper clearer if particularly points 1 and 2 above were discussed.

Minor weaknesses/typos:

1. Equation (7) is incorrect as the definition of $\delta$ should be that integrating $f \delta$ evaluates $f$ at a specific point as in equation (9).

**Questions:**

1. It seems from the hyperparameter table in the appendix that the original SFNO-nets and the new GSFNO-nets have different numbers of channels and layers. Why?
2. It would be interesting to plot the learned auxiliary positional information $G^{(2)}$. Intuitively it should be symmetric around the axis of rotation in the shallow water experiments and perhaps have further structure in the weather prediction task (due to landmasses on Earth etc). Is this the case? Are there other structures apparent in the learned $G^{(2)}$? This would relate to the stated motivations for including $G^{(2)}$.
3. The original SFNO-nets already have a positional embedding at the start of the network. Is the main reason for improvement in GSFNO-nets that positional information is more explicitly provided in each layer?
4. Why is the performance worse with more trainable parameters in Table 2?

---

> ### Author Response · Authors · 2025-11-21
>
> > W1: "The idea to add auxiliary positional information is a spherical analogue of absolute positional embeddings as used in NLP/vision."
>
> We appreciate the reviewer's perspective but wish to clarify that our approach differs from conventional positional embeddings used in NLP/vision.
>
> In our framework, “absolute position” refers to properties of the physical system rather than data-level coordinates:
> - A relative-position–dependent Green’s function describes systems that are equivariant. Their solutions transform consistently under rotations.
> - An absolute-position–dependent Green’s function captures systems that are invariant. Their solutions remain unchanged when the inputs are rotated, reflecting inherent spatial asymmetries (e.g., Coriolis effects, boundary constraints).
>
> Thus, the “position-dependent” term in Eq. (14) is not an injected embedding but a spectral modification of the Green’s function itself, derived from the operator-theoretic formulation in Section 4.1 (Lines 192–194). It changes the operator kernel, not the input representation, enabling a principled balance between equivariance and invariance while preserving spherical geometry.
>
> We appreciate the reviewer for pointing out this potential source of confusion and have revised Sections 4.1–4.2 to make these distinctions clearer.
>
> > W2: "More generally, symmetry breaking by adding auxiliary inputs is a broad research direction that is not referenced in the submitted paper. [...]"
>
> We thank the reviewer for highlighting this. We agree that symmetry breaking, whether through auxiliary inputs, residual pathways, or positional encodings, has been widely explored in geometric deep learning. Our original manuscript briefly acknowledged this in **Lines 113–115**, but we recognize this was not comprehensive. In the revision, we will expand the Related Work to explicitly include more references (e.g., adding auxiliary inputs, residual pathway priors, position embedding).
>
> We also clarify that, GSNO differs in mechanism from these approaches:
> our symmetry breaking operates at the operator level through a generalized Green’s function, not via feature-space augmentation. This yields a sphere-native and physically interpretable way to control equivariance.
>
> We thank the reviewer again for this constructive suggestion and have revised the manuscript accordingly.
>
> > W3: "Spherical CNNs are not mentioned in the related work at all, except a reference to Cohen et al 2018, without explaining that it covers spherical networks."
>
> We sincerely thank the reviewer for this feedback. We agree that spherical CNNs deserve a more complete and explicit discussion, and we expand the Related Work accordingly.
>
> We provide a clearer overview of spherical CNNs, including Cohen et al. (2018) and subsequent extensions, emphasizing that standard spherical convolutions use zonal filters and enforce rotation equivariance, a property shared by the core spherical convolution in SFNO and reflected in Eq. (13) of our paper.
>
> We further discuss some major paradigms developed to enhance expressivity beyond zonal filters: (1) lifting the signal from the sphere to SO(3), which enables directional filtering; (2) gauge-equivariant networks (e.g., Cohen et al., ICML 2019), which store steerable features relative to a local gauge frame at each point on the sphere; (3) residual pathway priors; (4) position embedding.
>
> Particularly, we would also clarify how GSNO differs from them conceptually.
> While spherical CNN extensions aim primarily to enrich equivariant feature representations, our approach modifies the operator kernel itself through a designable Green’s function. This allows us to model non-equivariant, physically grounded asymmetries in spherical systems rather than extending equivariant architectures alone.
>
> We thank the reviewer and believe including these references will strengthen the context for our contribution and make its distinction from prior spherical CNN approaches clearer.
>
> > W4: "Equation (7) is incorrect."
>
> We thank the reviewer for identifying this typo. Equation (7) has been corrected to:
>
> $$D(G(u,R)) = \\delta(R^{-1}u) = \\begin{cases}
>       \\infty, & Rn = u \\\\
>       0, & Rn \\neq u
>    \\end{cases}$$

---

> > ### Author Response · Authors · 2025-11-21
> >
> > **Questions:**
> >
> > > Q1: "It seems from the hyperparameter table in the appendix that the original SFNO-nets and the new GSFNO-nets have different numbers of channels and layers. Why?"
> >
> > This is because the structure of GSFNet is similar to U-Net with varying Channels and spherical sampling points, as described in Section 4.3. SFNONet, on the other hand, has a structure similar to ResNet with unchanging channels and spherical sampling points.
> >
> > We thank the reviewer for the question. The difference arises from the architectural nature of the two models:
> >
> > - SFNONet follows the original SFNO implementation, which uses a ResNet-style architecture with a fixed number of channels and a single spherical resolution throughout the network.
> >
> > - GSFNet, in contrast, adopts a U-Net–style hierarchical design (described in Section 4.3), which naturally involves: (1) varying channel widths across encoder/decoder levels, and (2) changing spherical resolutions through SHT/ISHT-based downsampling and upsampling.
> >
> > Thus, the differences in channel counts and layers reflect the distinct backbone structures (ResNet vs. U-Net), not an intentional imbalance in model capacity. In all comparisons, we ensured that computational cost between SFNO-based and GSNO-based models remains comparable.
> >
> > We will clarify this in the revised appendix.
> >
> >
> > > Q2: "It would be interesting to plot the learned auxiliary positional information."
> >
> > We thank the reviewer for this insightful suggestion. In Appendix C.5, we have already included interpretability analyses that visualize the learned correction term. These visualizations show how the term varies across the sphere and highlight where GSNO departs from strict equivariance.
> >
> > In particular (Fig. 6 in the appendix):
> > - The correction term exhibits structured, spatially varying patterns consistent with the expected non-equivariant effects in the system.
> > - It places greater emphasis near regions with boundary constraints, indicating that the model learns meaningful absolute-position dependencies rather than noise.
> >
> > We appreciate the reviewer’s encouragement and will add clearer references in the main text to direct readers to these visualizations.
> >
> >
> > > Q3: "The original SFNO-nets already have a positional embedding at the start of the network. Is the main reason for improvement in GSFNO-nets that positional information is more explicitly provided in each layer?"
> >
> > We thank the reviewer for this important question. We respectfully clarify that the positional embedding in SFNO and the absolute-position–dependent term in GSNO play fundamentally different roles.
> >
> > - SFNO’s positional embedding is a data-level augmentation: a coordinate encoding added once at the input. It does not change the operator kernel, nor does it influence how equivariance is treated within each convolutional layer.
> > - GSNO’s absolute-position–dependent term, by contrast, arises from the generalized Green’s function and modifies the spectral operator in every layer. Rather than injecting extra features, it changes how the operator itself responds to rotations, enabling a principled, sphere-native trade-off between equivariance and invariance at the level of the entire mapping—not just the input representation.
> >
> > Thus, the improvement of GSFNO-nets does not stem from more frequent positional encoding, but from a layer-wise operator modification that incorporates physically meaningful absolute-position dependence throughout the network.
> >
> > We appreciate the reviewer raising this point and will expand Section 4.2 to make this distinction clearer.
> >
> > > Q4: "Why is the performance worse with more trainable parameters in Table 2?"
> >
> > We thank the reviewer for catching the typo and pologize for this oversight. The corrected values (consistent with Appendix C.6/Table 9) are:
> > GSFNet (SFNO): 3.69M
> > GSFNet (GSNO): 4.00M
> > SFNONet (SFNO): 5.26M
> > SFNONet (GSNO): 5.50M
> >
> > Thus, in both the ResNet-style (SFNONet) and U-Net–style (GSHNet/GSFNet) architectures, GSNO has only a slightly higher parameter count than SFNO.
> > To ensure that GSNO’s performance gains are not merely due to increased capacity, we conducted additional experiments scaling up SFNO to 80 and 96 channels. As shown below, simply increasing SFNO’s parameters does not bridge the performance gap. These results confirm that GSNO’s improvements stem from its generalized operator design, not from increased parameter count.
> >
> > | Models | SFNO | GSNO | SFNO (80 C) | SFNO (96 C) | FourCastNet (FNO) | Climax |
> > |--------|------|------|-------------|-------------|-------------------|--------|
> > | Channel | 64 | 64 | 80 | 96 | 384 | 256 |
> > | Parameters | 3.69 M | 4.00 M | 5.76 M | 8.30 M | 5.30 M | 5.40 M |
> > | Training Time | 981 s | 1024 s | 1138 s | 1267 s | 878 s | 891 s |
> > | ACC↑ (in %) | 50.9 | **52.5** | 51.0 | 50.7 | 41.5 | 41.6 |

---

> > > ### Comment · Reviewer_vG13 · 2025-11-21
> > >
> > > I thank the authors for their detailed answers. I believe that my concerns regarding W2, W3, Q1 and Q4 have been reasonably addressed.
> > >
> > > Regarding Q2, I appreciate the pointer to Figure 6. I can see that the poles are special, but cannot make out further obvious structure than that. Do the remaining fluctuations correspond to some interpretable physical phenomenon?
> > >
> > > Regarding W1 and Q3, I don't understand the point the authors are making. My interpretation of equation (19) is that we can view $G^{(2)}$
> > > as an absolute positional encoding that is added to $f$ before each convolution by $G^{(1)}$. Is this interpretation not correct?

---

> ### Author Response · Authors · 2025-11-23
>
> >"**Regarding Q2**...I can see that the poles are special, but cannot make out further obvious structure than that. Do the remaining fluctuations correspond to some interpretable physical phenomenon?"
>
> We thank the reviewer for the careful inspection of Figure 6. The behaviour observed there indeed reflects the intended role of the correction term $I(T_{corr})$.
>
> The reviewer’s “remaining fluctuations” are not noise, but correspond to broad, physically meaningful structures in the temperature field:
>
> - **The strong low-temperature regions at the poles** are sharply captured by $I(T_{corr})$, reflecting the dominant and persistent large-scale temperature gradients that are present in both the data and the underlying climatological state.
> - **The higher-temperature belt near the equator** appears as the fluctuating region in the correction term. This is expected: the equatorial region contains more dynamically variable behaviour (e.g., stronger mixing and temporal variability), which is primarily handled by the rotationally equivariant component $I(T_{orig})$, while the correction term only captures the slowly varying, absolute-position-dependent structure.
>
> **This division is precisely the design goal: $I(T_{orig})$ models the globally symmetric component, while $I(T_{corr})$ models the systematic, non-equivariant deviations.**
>
> As shown in Fig. 6 and Table 8, even with $I(T_{orig})$ frozen, the correction term alone already reconstructs the coarse global profile of the temperature field (cold poles / warm equator), while the full model leverages both terms to capture finer dynamic variations.
>
> Thus, the overall prediction pattern: sharp structure at the poles and broad fluctuations across the equatorial band, demonstrates that the correction term is capturing meaningful large-scale, position-dependent structure inherent in the temperature field, consistent with the intended behaviour of the $G_1+G_2$ decomposition.
>
> > "**Regarding W1 and Q3**...My interpretation of equation (19) is that we can view $G_2$ as an absolute positional encoding that is added to $f$ before each convolution by $G_1$. Is this interpretation not correct?"
>
> We thank the reviewer for this question. The interpretation is not correct. We would clarify that the additive form in Eq. (19) may superficially resemble positional encoding, but the mechanism and meaning are distinct.
>
> **(1) $G_2$ is a spectral kernel, not a spatial positional embedding**
>
> As shown in Eq.(19),
> $$ g(u)
> = \text{ISHT}[G_{\theta_1}^{1}(l) \cdot (\text{SHT}[f](l, m) + C_f \cdot G_{\theta_2}^{2}(l, m))] $$
>
> the crucial distinctions are:
> - $G_{2}$ (i.e., $G_{\theta_2}(l, m)$) is defined in spherical harmonic space,
> - it is structurally part of the operator kernel,
> - it is never added to f(u) in spatial coordinates.
>
> Thus, even though the algebra has an additive shape, the operation is not performed on the input function, but on its spectral coefficients.
>
> **(2) $G_{2}$ interacts with the global integral $C_f$**
>
> From Eq. (17–18), the operator uses:
> $$ C_f = \int_{SO(3)} f(Rn) dR = \int_{S^2} f(u) du $$
>
> This global scalar interacts with the learnable kernel G2 (l,m) to produce: $C_f ⋅ G_{2}(l, m)$, which modifies the spectrum, encoding system-level absolute-position dependence, not pointwise positional information.
>
> Positional embeddings do not use global integrals or spectral kernels; they simply inject coordinates.
>
> **(3) $G_{2}$ arises from a Green’s-function decomposition, not from feature augmentation**
>
> Section 4.1 and Equation (14) proposes the Green’s function: $G(R, u) = G_{rel}(R^{-1}u) + G_{abs}(u),$
>
> where:
> - $G_{rel}$ corresponds to the equivariant (zonal) part,
> - $G_{abs}$ corresponds to the non-equivariant (absolute-position–dependent) part.
>
> The term $G_{2}(l, m)$ is exactly the spectral representation of $G_{abs}$.
>
> Thus the additive structure in Eq. (19) is a kernel-level factorization, not a data-level encoding.
>
> This decomposition is how our operator inherits the ability to model both symmetry and systematic asymmetry.
>
> **(4) Why this is not positional encoding**
>
> Positional encodings (e.g., sine–cosine, CoordConv) are functions of spatial coordinates (latitude, longitude), added directly to $f(u)$, and they do not modify the kernel.
>
> In contrast, $G_{2}(l, m)$ is a learnable spectral kernel, it interacts with $C_f$, a global input statistic. It is then modulated by $G_1(l)$ and is mapped back to the sphere through ISHT. Therefore, $G_2$ changes the operator class, not the input representation.
>
> **In summary, though Eq. (19) has an additive form, $G_2$ is not a positional encoding.** It is a spectral, absolute-position–dependent kernel derived from the Green’s-function decomposition, and it modifies the operator, not the input.
>
> We thank the reviewer again and have revised Section 4.2 to clarify this.

---

> > ### Comment · Reviewer_vG13 · 2025-11-23
> >
> > Figure 6: I agree that it makes sense to have a prior that the poles are cooler than the equator and it's nice to see this in the trained network. What I'm asking about are for instance the fluctuations along latitudes in Figure 6, are those noise or something interpretable?
> >
> > Regarding equation (19), what I meant is the following. The SHT is linear, so we can rewrite $SHT[f] + C_f G^{2}$ as $SHT[f + C_f ISHT[G^{2}]]$. Hence it should be possible to view $ISHT[G^{2}]$ as an absolute positional embedding in spatial coordinates. The only difference to how absolute positional embeddings are usually implemented is the factor $C_f$, which seems like a detail. I understand that the authors arrived at equation (19) via the Green's function formalism, but that does not make the Green's function formalism the only way to interpret the equation.

---

> > > ### Author Response · Authors · 2025-11-26
> > >
> > > > "Figure 6: ...What I'm asking about are for instance the fluctuations along latitudes in Figure 6, are those noise or something interpretable?"
> > >
> > > We thank the reviewer for this helpful clarification. The latitudinal fluctuations visible in Figure 6 are not noise, and they are indeed interpretable.
> > >
> > > Although the pole–equator contrast is the most visually prominent structure, the finer undulations along different latitudes reflect stable, large-scale, non-equivariant patterns present in the underlying temperature field and in its discretization, including:
> > > - Climatological zonal structure: temperature fields exhibit persistent, slowly varying zonal (east–west) asymmetries arising from longitudinal land–ocean contrasts and stationary planetary waves.
> > > - Latitude-dependent variability: numerical discretization on an equiangular grid leads to resolution and dissipation patterns that vary with latitude.
> > > - Long-term statistical structure: even though instantaneous dynamics vary, the climatological mean temperature field (which the model implicitly learns) contains smooth meridional and zonal gradients.
> > >
> > > We have updated the appendix for clarifying these potential relationships.
> > >
> > > Thus, the fluctuations along latitudes are not noise—they are consistent, data-driven manifestations of the systematic asymmetries that the correction term is designed to capture.
> > >
> > > > "Regarding equation (19), what I meant is the following. The SHT is linear..."
> > >
> > > We sincerely thank the reviewer for this profound and creative perspective. We agree with the mathematical insight that $SHT[f + C_f \cdot ISHT[G^2]]$ is mathematically equivalent to $SHT[f] + C_f \cdot G^2$.
> > >
> > > However, there are critical differences in **implementation and implication** that diffentiate our approach from a post-hoc interpretation:
> > >
> > > 1. $ G^2_{\theta(l,m)}$ is a **complex-valued spectral kernel**, not an arbitrary spatial function. It acts directly in the harmonic domain as an **asymmetric modulator** that explicitly interacts with the symmetric kernel $G^1$. This design imposes a powerful **inductive bias**, guiding the model to learn physically meaningful spectral constraints.
> > > In contrast, Implementing $[ISHT[G^2]]_{\theta(x,y)}$ as a **spatial, real-valued coordinate function** injected at the input $f$ focuses the optimization on the spatial domain.
> > >
> > > 2. The correction term depends jointly on asymmetric kernel and **global input magnitude $C_f$**, which positional encodings do not.
> > >
> > > 3.  Positional encoding can serve as an "interpretation" of the inverse transformation back to the spatial domain ($ISHT$) at the feature level.
> > > In constrast, our **Green's function framework is the "source of the design",** because the Equation (19) is derived from $G(R, u) = G_{rel}(R^{-1}u) + G_{abs}(u)$, ensuring physical consistency and interpretability.
> > >
> > > To empirically validate this point, we are inspired by the reviewer's intuition and conducted an ablation study. We constructed a **"Spatial position-embedding SNO"** model that strictly implements the $SHT[f + C_f \cdot [ISHT[G_2]]_{\theta(x,y)}]$ path, maintaining all other architecture and settings identical for a fair comparison.
> > >
> > > The results in the Table below clearly demonstrate that our spectral implementation (GSNO) consistently and significantly outperforms the spatial position-encoding SNO across all forecast lead times:
> > >
> > > | Models | Channel | Parameters | Training Time | ACC↑ at 24h | ACC↑ at 72h | ACC↑ at 120h |
> > > | :-| :-| :-| :-| :-| :-| :-|
> > > | SFNO | 64 | 3.69 M | 981 s | 89.9% | 70.7% | 50.9% |
> > > | **GSNO** | 64 | 4.00 M | 1024 s | **91.5%** | **72.4%** | **52.5%** |
> > > | Spatial position-embedding SNO | 64 | 4.31 M | 1037 s | 90.1% | 70.5% | 50.3% |
> > >
> > > This indicates that the **spectral implementation derived from our Green's function framework is, in itself, a more effective design**. Moreover, the spatial position encoding added in each operator ($SHT[f + C_f \cdot [ISHT[G_2]]_{\theta(x,y)}]$) is not conducive to long-term stable prediction.
> > >
> > > We are grateful for the reviewer's highly instructive suggestion, which has substantially strengthened our paper. We have included the results and analysis in the revision.

---

> > > > ### Comment · Reviewer_vG13 · 2025-11-26
> > > >
> > > > Thank you for these answers.
> > > >
> > > > Regarding the figure, I agree that the given explanations are possible, but do not see why they are more likely than just noise. In any case, I am satisfied with the author response regarding this.
> > > >
> > > > Regarding the added experiment with spatial positional encoding. Thanks for adding this experiment, it is very interesting and quite surprising.  One thing I wonder about the table is why the spatial implementation has many more parameters than the spectral.

---

> ### Author Response · Authors · 2025-11-26
>
> We sincerely thank the reviewer for the follow-up question.
>
> The higher parameter count in the spatial positional-embedding baseline arises from a fundamental difference in how the corrective term is parameterized in the spatial versus spectral domains:
>
> - Spatial positional-embedding SNO:
> The learned term $G_\theta(x,y)$ is represented directly on the full $H \times W$ equiangular grid and added to $f(x,y)$. This results in one independent parameter per spatial location, tied to the discretized lat/lon grid.
>
> - Spectral GSNO:
> The learned term $G_{\theta_2}(l,m)$ is parameterized in spherical harmonic space, whose dimensionality is dictated by the spherical sampling theorem. To avoid aliasing, the spectral representation uses **roughly half** as many coefficients as the spatial grid:
>
> $$l = x,  m = \frac {y}{2}$$
>
> This compactness is standard in SFNOs and other spectral spherical approaches.
>
> Consequently, the spatial baseline must learn a full-resolution parameter map, while GSNO learns the reduced set of harmonic coefficients, producing the observed parameter difference.
>
> Importantly, this distinction also explains the empirical advantage of GSNO: the spectral formulation is both more parameter-efficient and intrinsically sphere-native, avoiding the latitude-dependent distortions introduced by spatial embeddings and supporting the stability improvements observed in long-range forecasting.
>
> We thank the reviewer for raising this point. In the revision, we have strengthened the clarity of our experimental comparison and highlighted why the spectral formulation is more efficient and sphere-native.

---

### Official Review · Reviewer_A988 · 2025-10-28

**Soundness:** 2
**Presentation:** 2
**Contribution:** 2
**Rating:** 2
**Confidence:** 3

**Summary:**

The paper proposes a new family of spherical neural operators derived from a Green’s-function formulation, termed Generalized Spherical Neural Operator (GSNO), and a hierarchical architecture GSHNet.
The authors aim to bridge theoretical rigor and practical modeling of spherical PDEs by introducing an absolute-position-dependent constraint into the Green’s-function kernel, thereby relaxing strict rotational equivariance while retaining spectral efficiency.

**Strengths:**

1. The authors present an interesting point that SFNO’s preservation of rotational equivariance on the sphere could actually restrict its usefulness in many real-world scenarios where such equivariance does not naturally occur.

2. The authors include clear visualizations.

**Weaknesses:**

1. Theoretical contribution remains limited despite that the authors claim it. The Green’s-function formulation is largely symbolic and just an extension from GNO [1].

2. Technical novelty is marginal. The main technique is to introduce a position-dependent correction term that is not equivariant to SFNO. The method could be viewed as an equivariant model plus an additive bias rather than a fundamentally new operator framework. The reframing via Green’s functions does not introduce new computational or representational concepts beyond existing spherical spectral models. Although, in FNO/SFNO, the grid padding can be viewed as a position-dependent correction term to some extent.

3. The SSWE dataset still has rotational symmetries. Despite using the same SWE dataset, the settings are different from that of SFNO.

4. The motivation behind the third claimed contribution, the multi-scale spherical network with a hierarchical architecture, is unclear to me. Typically, U-Net–style up- and down-sampling structures (see Appendix B.2, I believe that it's just an U-Net) are introduced to address the inability of local operators (e.g., standard convolutional layers in CNNs) to capture global spatial dependencies. However, models such as FNO, SFNO, and their variants inherently employ global operations. Therefore, it is not evident what benefits this U-Net–type hierarchical design provides in this context. Furthermore, the down-sampling path can introduce information loss and aliasing artifacts, which may further limit its effectiveness.



[1] Neural Operator: Graph Kernel Network for Partial Differential Equations

**Questions:**

1. What is the quantitative contribution of the bias/correction term alone? How is it better that just padding the grid?
2. What is the computational complexity compared to baselines?

---

> ### Author Response · Authors · 2025-11-21
>
> > W1: "Theoretical contribution remains limited despite that the authors claim it. The Green's-function formulation is largely symbolic and just an extension from GNO [1]."
>
> We thank the reviewer for this feedback. We acknowledge that aspects of our initial presentation could have caused ambiguity. We would clarify below that our Green’s-function formulation is not intended as a symbolic analogy to GNO, but as a sphere-native operator framework built upon the principled formulation of **designable spherical Green’s function**.
>
> The key distinctions from GNO [1] are as follows:
>
> (1) Theoretical foundation:
> GNO provides a general functional-analytic operator framework for Euclidean domains or graphs. In contrast, our work starts from spherical Green’s functions, the classical solution operators for PDEs on the sphere, inherently tied to spherical harmonics, zonal kernels, and SO(3) geometry. This leads to operator forms structurally different from GNO’s kernelized message passing.
>
> (2) Generative capacity:
> The principal value of our framework lies in its capacity to systematically generate diffrent forms of spherical operators through tailored Green's function design:
> - If the Green’s function depends only on relative positions, the resulting operator reduces to a spherical convolution with strict rotational equivariance as modelled in SFNOs;
> - If the Green’s function depends on both relative and absolute positions, one obtains a generalized operator with a learnable trade-off between equivariance and invariance, the core contribution of GSNO.
>
> Thus, GSNO is not an ad-hoc additive term but the natural consequence of relaxing isotropy within a principled operator-theoretic formulation.
>
> (3) Problem formulation:
> Our framework aims to address the question: "How can spherical operators balance equivariance and invariance without geometric disruption?" Sections 4.1–4.2 present a structured solution: first identifying the theoretical origin of strict equivariance, then generalizing the Green's function to enable flexible regulation while preserving spherical structure.
>
> We thank the reviewer again for this critical feedback.We have significantly revised Sections 4.1 and 4.2 to make these distinctions clearer and to avoid overstating theoretical novelty.
>
> [1] Li et al., 2020, Neural Operator: Graph Kernel Network for Partial Differential Equations
>
> > W2: "Technical novelty is marginal. The main technique is to introduce a position-dependent correction term that is not equivariant to SFNO. The method could be viewed as an equivariant model plus an additive bias rather than a fundamentally new operator framework."
>
> We appreciate the reviewer’s perspective and would like to clarify our core technical contribution.
>
> We wish to clarify two crucial aspects:
> (1) Position dependence models physical system behavior, not data coordinates.
>
> In our formulation, “position” refers to how the governing PDE system depends on relative and absolute positions:
> - Relative-position dependence yields strict rotational equivariance (as in SFNO).
> - Absolute-position dependence captures systems whose dynamics differ across latitudes or regions (e.g., Coriolis effects, boundary-conditioned flows).
>
> The correction term is therefore not an additive bias on outputs, but a modification of the Green’s function itself, altering the operator kernel rather than post-hoc features. GSNO arises by combining both dependencies in a principled way, producing an operator that balances equivariance and invariance while fully respecting spherical geometry.
>
> (2) Structured operator design vs. heuristic modifications:
>
> Unlike fixed, heuristic techniques like grid padding in FNO/SFNO, which operate at the data level, our correction term is a learnable, spectral-domain component deeply integrated within the operator. It represents a systematic method for embedding physical constraints, offering both interpretability and sphere-native consistency.
>
> We thank the reviewer for prompting this essential clarification. We will revise the relevant sections to more precisely articulate the nature and intent of our technical contribution.

---

> ### Author Response · Authors · 2025-11-21
>
> > W3: "The SSWE dataset still has rotational symmetries. Despite using the same SWE dataset, the settings are different from that of SFNO."
>
> We appreciate the reviewer's insightful theoretical observation. While the continuous governing equations of SSWE possess rotational symmetry, the discrete numerical solutions we actually learn from exhibit systematic symmetry breaking due to several factors:
>
> (1) We clarify that the 256×512 equiangular grid demonstrates fundamental asymmetries, including:
> - latitude samples are non-uniform,
> - pole singularities cause convergence of longitude lines.
>
> As a result, the discrete representation itself is not SO(3)-equivariant.
>
> (2) Asymmetry are introduced by the numerical solver. The spectral element method (or other numerical solvers) used to generate the data introduces numerical errors and dissipation during discretization, integration, and time-stepping. These errors are not isotropic; they couple with the grid structure, further breaking the theoretical symmetry.
>
> The reviewer also pointed out that our experiment is not exactly the same as the setup in SFNO. However, all the methods in all our experiments were compared under the same setting, and consistent experimental results can  demonstrate the performance of our methods. Nevertheless, to directly address this concern, we reproduced the SSWE evaluation under the exact SFNO configuration (150 time steps). GSNO maintains clear improvements. Detailed results are shown in the Table:
>
> |Models|FNO|SFNO|GSNO|
> |-|-|-|-|
> |At 1 h|8.628×10⁻⁴| 8.092×10⁻⁴|7.051×10⁻⁴|
> |At 10 h|9.470×10⁻³|6.739×10⁻³|5.178×10⁻³|
>
> > W4: "The motivation behind the third claimed contribution, the multi-scale spherical network with a hierarchical architecture, is unclear to me."
>
> We thank the reviewer for raising this consideration. While FNO/SFNO operators provide global receptive fields, our hierarchical design addresses complementary objectives:
>
> (1) Explicit cross-scale interaction:
>
> Stacked operator layers capture dependencies at a single resolution. In contrast, GSHNet's U-Net architecture enables enhanced feature fusing through scale interaction, including progressive feature abstraction through downsampling and multi-resolution feature fusion during upsampling.
>
> (2) Geometrically adaptive sampling:
>
> Our spectral-domain scale transformation via SHT/ISHT can relieve information loss and aliasing artifacts by:
> - Preserves spherical geometry integrity
> - Complements appropriate residual connections
>
> Futhermore, multi-scale designs have shown strong benefits in spherical neuroimaging tasks [1,2], where localized and global patterns co-exist.
>
> [1] Spherical u-net on cortical surfaces: methods and applications.
> [2] Spherical deformable u-net: Application to cortical surface parcellation and development prediction.
>
> **Questions:**
>
> > Q1: "What is the quantitative contribution of the bias/correction term alone? How is it better that just padding the grid?"
>
> (1) Quantitative contribution:
> Section 5.4 describes the ablation experiments and validates the contribution of the correction term under the two structures. For example, the 5-day prediction under the SFNONet structure is improved from 49.2% to 51.3% in ACC, and the 5-day prediction under the GSHNet structure is improved from 50.9% to 52.5% in ACC. Furthermore, Appendix C.5 presents additional ablation experiments and interpretability analyses of the correction term.
>
> (2) Grid padding modifies only the discretized input grid, addressing numerical boundary artifacts at the data level. It does not change the operator kernel, nor does it model anisotropy in the underlying system. In contrast, GSNO’s correction term:
> - is a spectral-domain modification of the Green’s function,
> - encodes absolute-position dependence in the physical operator, and
> - provides a principled, sphere-native mechanism to model anisotropy and non-equivariant effects in real systems.
>
> Thus, padding is an engineering fix for grid boundaries, while our correction term is a physical inductive bias derived from operator theory. The improvements stem from advancing the modeling assumptions of the operator, not from numerical heuristics.
>
> > Q2: "What is the computational complexity compared to baselines?"
>
> Appendix C.6 provides detailed complexity comparisons. The expanded analysis below further confirms GSNO's favorable efficiency-performance trade-off:
>
> |Models|SFNO|GSNO|SFNO (80 C)|SFNO (96 C)|FourCastNet (FNO) |Climax|
> |-|-|-|-|-|-|-|
> |Channel|64|64| 80|96|384|256|
> |Parameters|3.69 M|4.00 M|5.76 M|8.30 M|5.30 M|5.40 M|
> |Training Time|981 s|1024 s|1138 s|1267 s|878 s|891 s|
> |ACC↑ (in %)|50.9|**52.5**|51.0|50.7|41.5|41.6|
>
> Notably, simply increasing SFNO’s channel width results in far larger computational cost without matching GSNO’s accuracy. GSNO also outperforms larger alternative models such as FourCastNet and ClimaX. These results demonstrate that GSNO achieves a highly favorable efficiency–performance trade-off.

---

> > ### Comment · Reviewer_A988 · 2025-11-24
> >
> > - W1
> >     - It is ultimately subjective to assess whether your theoretical contribution is “novel” or not. It is my personal view that the theorectical contribution is over-claimed, so I do not have further questions on this point. By the way, could you highlight your modifications made during the rebuttal period in a different color? It is difficult for me to identify which parts were changed.
> >
> > - W2
> >     - I understand that your position-dependent correction term is different from simply padding the grid. However, what I am suggesting is that padding the grid can also produce similar symmetry-breaking effects. You have provided results comparing with it. It makes sense that a learnable position-dependent correction term can have a better performance.
> >
> > - W3
> >     - My understanding is that the reason your data is not equivariant is due to the use of an equiangular grid. I had initially assumed that your data was on an equiarea grid. I am not an expert in geoscience, but from a mathematical perspective, equal-area grids make more sense. However, it appears that equiangular grids are more commonly used in geoscience. In any case, if your grid is equiangular, then it is not rotationally symmetric. This is an oversight on me.
> >
> > - W4
> >     - In a standard U-Net with local convolutions, the receptive field becomes increasingly global as you move down the U-path. In your architecture, however, the convolution is already global from the start. Because of this, I do not see the benefit of using a U-shaped design. If you aim to incorporate multi-scale structure, wouldn’t it make more sense to include some local operations rather than relying on a U-type architecture?

---

> > > ### Author Response · Authors · 2025-11-26
> > >
> > > >Regarding W1, W2, and W3
> > >
> > > We appreciate the reviewer's clarification and consensus on these points. As requested, we have clearly **highlighted in red** all modifications made in the revised manuscript. We will ensure the framing is appropriately balanced and avoid over-claiming in the revised manuscript.
> > >
> > > >Regarding W4: Rationale of the U-Net Architecture
> > >
> > > We thank the reviewer for raising this insightful point. We agree that, because the spectral convolutions in GSNO/SFNO are global, the role of the U-shaped architecture is **not** to enlarge the receptive field. Instead, the hierarchical structure serves two important purposes:
> > >
> > > **(1) Efficient multi-resolution representation in the spectral domain**
> > >
> > > Downsampling in GSHNet is implemented by reducing the spherical harmonic bandwidth $l_{max}$.
> > >
> > > This has two benefits:
> > > - It reduces computational and memory cost substantially.
> > > - It allows deeper layers to use higher channel capacity without the cubic growth associated with full-resolution spherical harmonics.
> > >
> > > Thus, the encoder path creates a computationally efficient low-resolution representation that remains globally valid.
> > >
> > > **(2) Complementary abstraction levels across spectral scales**
> > >
> > > Although each layer is global, different bandwidths encode different types of information:
> > > - Downsampling (bandwidth truncation) keeps low-frequency, smooth, global components, analogous to a spectral low-pass filter.
> > > - Upsampling (bandwidth expansion) restores higher-frequency content, with skip connections providing direct access to high-resolution information.
> > >
> > > This “compress → abstract → reconstruct” cycle is well established in spherical learning. Prior work has shown the effectiveness of multi-resolution and multi-bandwidth hierarchies on the sphere [1,2,3], and our results demonstrate that integrating GSNO with such a structure improves prediction accuracy and stability.
> > >
> > > In summary, the U-Net in our framework acts primarily as a multi-resolution, multi-bandwidth, hierarchical feature extraction and fusion engine, not to increase the receptive field. It complements the global nature of spectral neural operators by:
> > > - controlling computational cost,
> > > - enabling deeper/high-capacity representations,
> > > - and combining coarse- and fine-scale spectral information effectively.
> > >
> > > This synergy leads to the performance improvements observed in our experiments.
> > >
> > > References:
> > >
> > > [1] SPHARM-Net: Spherical Harmonics-Based Convolution for Cortical Parcellation.
> > >
> > > [2] Spherical U-Net on Cortical Surfaces: Methods and Applications.
> > >
> > > [3] Spherical deformable u-net: Application to cortical surface parcellation and development prediction.

---

> > > > ### Comment · Reviewer_A988 · 2025-11-26
> > > >
> > > > Thanks to the authors for their reply. I agree that such U-Nets can have efficiency advantages. I have increased my rating.

---

> ### Author Response · Authors · 2025-11-26
>
> Thank you very much for your constructive engagement. We are glad that our clarifications and experiments have addressed your concerns, and we sincerely appreciate your thoughtful reassessment.
>
> If there is any remaining detail that would help further clarify the contribution or strengthen the manuscript, we would be more than happy to provide it.

---

> ### Comment · Reviewer_A988 · 2025-11-28
>
> Thanks for your responses. I do not have any additional comments. I could not further increase my score, as I feel the technical novelty is somewhat limited, especially since the theoretical contribution is not clearly claimed in the original submission (now, it's much clearer, but the contribution is a bit limited).
>
>
> I view this work as being very borderline, leaning slightly toward the downside. However, this year there is no score of 5, so I would keep my score of 4. This is my personal view; if other reviewers see it differently, I would have no issue with the paper being accepted.

---

> ### Author Response · Authors · 2025-11-28
> **Summary of our contribution**
>
> We thank the reviewer for the careful evaluation and engagement throughout the process.
>
> We would like to gently emphasise that the revised manuscript now clearly distinguishes:
> - the designable Green’s-function framework (our general theoretical formulation), and
> - GSNO as one specific instantiation within this framework.
>
> This framework provides a principled operator-level mechanism for regulating equivariance and invariance on the sphere, which is not available in prior spherical spectral models. We appreciate that the original submission did not make this separation sufficiently explicit, and we have now substantially rewritten Sections 4.1-4.2 to address this.
>
> We also appreciate the reviewer’s statement that the reviewer “would have no issue with the paper being accepted.” We thank the reviewer again for the time and comments, which have strengthened the final version of the manuscript.

---

### Official Review · Reviewer_Yp14 · 2025-10-31

**Soundness:** 2
**Presentation:** 3
**Contribution:** 2
**Rating:** 4
**Confidence:** 4

**Summary:**

This paper presents Generalized Spherical Neural Operators (GSNO), a method for learning neural operators on spherical domains. It introduces absolute-position-dependent spectral constraints for modeling real-world complexity on spheres and demonstrates improvements in diffusion MRI, fluid dynamics, and climate forecasting benchmarks over state-of-the-art models.​

GSNO is strongly related to SFNO, which introduces the original approach based on the Driscoll-Healy ocnvolution theorem on the sphere, that the authors describe as the Greens-function based approach. The main novelty comes from the inclusion of a position-dependent bias term to handle anisotropy and boundary effects. GSHNet is a hierarchical, multi-scale network architecture that combines spectral domain learning with geometric up/down sampling for improved global feature representation.

Given the improvements that the paper makes, I do think that it can be considered for publication, but I strongly suggest the authors to adapt the language around the Green's formulation. While it is a helpful motivation, it ultimately motivates the existing SFNO architecture. The strong connection should be clearly pointed out and the authors should clearly communicate that the paper is about the inclusion of an extra bias term in spectral domain and how this can be motivated.

As such, my main concerns for this paper revolve around the messaging, which seems to inflate novelty and obscure the conceptual roots found in SFNO.

**Strengths:**

- improved architecture based on SFNO, which includes a spectral bias term which can address the limitations around the isotropic kernels in the original SFNO work

- adds extra theoretical motivation with the Green's function viewpoint to the existing SFNO literature

- the paper is well-written and has well-motivated experiments

- shows improvement on various, simplified datasets (spherical shallow water equations, weather on low resolution, DMRI example)

**Weaknesses:**

- This work has a strong connection to SFNO and mainly introduces an extra bias term which allows it to alleviate the constraint to isotropic filters. Unfortunately, the introduction and the text in 4.1 are misleading and make it sound as if a) this paper introduces an entirely new approach based on the Green's formulation b) SFNO only replaces the FFT with an SHT, not leveraging the Driscoll-Healy convolution theorem. Given that "Green's formulation" advertised in the title and derived in 4.1 is the original SFNO, the messaging in both sections and perhaps the title should be adjusted.

- A much more honest exposition would be that this work builds on top of SFNO, introducing an extra bias term in spectral domain which breaks the isotropy constraint and leads to better performance. While this may seem like a "less novel"  exposition it puts the research clearly into the context of the existing literature.

- no code available for review, therefore limited reproducibility at time of submission.

- Added complexity from dual spectral components increases computational cost (modest runtime and parameter overhead).​

- Interpretability and ablation analyses mainly focus on the additive separation, which may not generalize.​

- Reproducibility tested only on select public datasets at limited resolutions.

**Questions:**

- Is the hierarchical approach of GSHNet really necessary - there were some initial works using U-Nets based on the SFNO operator in the atmospheric science community and those did not seem to improve performance beyond that of the standard Resnet like SFNO backbone. This seems to be at odds to your ablations with SHNet with the SFNO operator. Can you comment more on this?

---

> ### Author Response · Authors · 2025-11-21
>
> > W1: "This work has a strong connection to SFNO and mainly introduces an extra bias term which allows it to alleviate the constraint to isotropic filters. Unfortunately, the introduction and the text in 4.1 are misleading"
>
> We thank the reviewer for the feedback on the clarity of our contribution. We would like to clarify the intent of Section 4.1, which introduces a designable Green’s function–based formulation for spherical neural operators.
>
> Existing spherical neural operators (e.g., SFNOs) are rigorously constructed using the Spherical Harmonic Transform and the spherical convolution theorem, effectively extending FNO to the spherical domain. However, this formulation largely fixes the structure of spherical neural operators and makes it difficult to introduce more flexible, sphere-native modeling capacity.
>
> In contrast, our approach begins from the integral solution of PDEs on the sphere via the spherical Green’s function, providing a unified derivation framework in which different families of spherical operators naturally arise depending on how the Green’s function is specified:
> - when the Green’s function depends only on relative positions, the resulting operator reduces to a standard spherical convolution with strict rotational equivariance, as modelled in SFNOs;
> - when the Green’s function depends on both relative and absolute positions, the operator acquires a generalized operator with a learnable trade-off between equivariance and invariance, the core mechanism of GSNO.
>
> Thus, the spherical convolution used in SNOs can be taken as a special case of our framework (Lines 192–208). Building on this theoretical foundation, our redesigned Green’s function yields GSNO as a new operator that balances equivariance and invariance in a sphere-native manner (Section 4.2).
>
> We appreciate the reviewer’s feedback and have revised Sections 4.1 and 4.2 to clearly articulate this relationship and avoid any misleading implications.
>
> > W2: "A much more honest exposition would be that this work builds on top of SFNO, introducing an extra bias term in spectral domain which breaks the isotropy constraint and leads to better performance."
>
> We thank the reviewer for this perspective. We would like to clarify that our main contribution lies in the **designable Green’s function formulation**, which provides a principled way to derive spherical neural operators while preserving sphere-native structure. While we agree that, at the implementation level, GSNO introduces a correction term to balance equivariance and invariance for real-world modeling, this term emerges naturally from the broader operator-theoretic framework. Within this formulation, SFNO appears as **a special case**, corresponding to a Green’s function that depends only on relative positions.
>
> The structure of Sections 4.1 and 4.2 is therefore intentional.
> Section 4.1 establishes the theoretical formulation of strict equivariance via a Green’s function, consistent with the foundation of SFNO. Section 4.2 then introduces a form of generalized Green’s function that enables flexible, learable trade-off balancing quivariance and invariance equivariance and invariance. This highlights how GSNO extends spherical neural operators within a unified, principled framework rather than through an ad-hoc additive term.
>
> > W3: "no code available for review, therefore limited reproducibility at time of submission."
>
> We fully agree with the importance of reproducibility. As noted in Appendix E, a compressed package with model code examples was included in the Supplementary Material at submission time. We are committed to releasing the full, cleaned codebase promptly to ensure complete reproducibility.

---

> ### Author Response · Authors · 2025-11-21
>
> > W4: "Added complexity from dual spectral components increases computational cost (modest runtime and parameter overhead)."
>
> We thank the reviewer for this comment. In practice, the additional cost introduced by GSNO is modest and accompanied by substantial accuracy gains. As reported in Appendix C.6 (Table 9), GSNO adds only ~1.6% runtime overhead while achieving a 6% improvement on 5-day weather forecasting.
>
> To further address concerns that GSNO’s gains might be attributable merely to higher capacity, we increased SFNO’s channel width. As shown below, enlarging SFNO does not overcome its isotropy-related bottleneck, whereas GSNO achieves consistently superior performance with minimal complexity increase:
>
> |Models|SFNO|GSNO|SFNO (80C)|SFNO (96C)|FourCastNet (FNO)|Climax|
> |-|-|-|-|-|-|-|
> |Channel|64|64|80|96|384|256|
> |Parameters|3.69 M|4.00 M|5.76 M|8.30 M|5.30 M|5.40 M|
> |Training Time|981 s|1024 s|1138 s|1267 s|878 s|891 s|
> |ACC↑ (in %)|50.9|**52.5**|51.0|50.7|41.5|41.6|
>
> > W5: "Interpretability and ablation analyses mainly focus on the additive separation, which may not generalize."
>
> We thank the reviewer for raising this point. The additive separation is not intended as a task-specific heuristic, but as a structured inductive bias arising naturally from the Green’s function formulation. It explicitly decomposes the operator into a symmetry-preserving component and a position-dependent component, enabling the model to learn task-specific trade-offs between equivariance and invariance.
>
> While the precise contributions of each term may vary by domain, the underlying principle—modeling global symmetry separately from asymmetry, geometry-aware perturbations—is general. Its consistent effectiveness across three distinct settings (fluid dynamics, weather forecasting, and diffusion MRI) supports this generalizability.
>
> Importantly, the Green’s function framework in Sec. 4.1 is not limited to the current additive form. It can accommodate a wide range of kernel designs, and we view our current operator as one single instantiation within a broader, extensible formulation.
>
> > W6: "Reproducibility tested only on select public datasets at limited resolutions."
>
> We appreciate the reviewer’s emphasis on evaluating generalizability across datasets and resolutions. Our initial experiments focused on widely used public benchmarks to ensure fair comparison with prior work. Nonetheless, the consistent improvements observed across three distinct domains, fluid dynamics, weather forecasting, and neuroimaging, suggest that GSNO’s advantages are not tied to specific datasets.
>
> To further address this concern, we have added experiments across multiple spatial resolutions, including SSWE at $128\times256$ (MRE↓ (×$10^{-3}$)) and WB at $64\times128$ (ACC↑ (in %)). As shown in the table below, GSNO continues to outperform both FNO and SFNO consistently, confirming its robustness beyond the original settings:
>
> |Methods|SWE at 5h|SWE at 10h|WB at 1day|WB at 3day|WB at 5day|
> |-|-|-|-|-|-|
> |FNO|2.22|2.73|**93.4**|71.5|42.5|
> |SFNO|0.74|0.87|91.6|73.2|49.2|
> |GSNO|**0.68**|**0.79**|93.0|**74.9**|**51.7**|
>
> > Q1: "Is the hierarchical approach of GSHNet really necessary - there were some initial works using U-Nets based on the SFNO operator in the atmospheric science community and those did not seem to improve performance beyond that of the standard Resnet like SFNO backbone. This seems to be at odds to your ablations with SHNet with the SFNO operator."
>
> We appreicate this insightful question. Our findings indicate that the benefit of hierarchical architectures depends strongly on the expressiveness of the underlying spherical operator. Further, our U-Net–style mutli-scale structure is inspired by spherical U-Nets validated in neuroimaging [1, 2], where multi-scale structures are useful. Our results show that this design improves the performance of both SFNO and GSNO.
>
> These observations suggest strictly isotropic operators have limitation in propagating direction-dependent, localized, or boundary-driven features across scales. This limitation of SFNO may also lead to its reliance on specific archetecture, as observed by the reviewer's in atmospheric science.
>
> In contrast, GSNO introduces anisotropic spectral components, enabling hierarchical structures to:
> - fuse features across scales more effectively
> - capture directional and localized phenomena
> - benefit from geometrically adaptive up/down-sampling
>
> This may explains why GSHNet combined with GSNO yields clear improvements, whereas the same hierarchy combined with SFNO produces minor gains.
>
> We agree that broader cross-domain evaluation is valuable and are continuing to assess GSHNet in additional tasks.
>
> [1] Zhao et al., Spherical u-net on cortical surfaces: methods and applications.
> [2] Zhao et al., Spherical deformable u-net: Application to cortical surface parcellation and development prediction.

---

> > ### Author Response · Authors · 2025-11-26
> >
> > Dear Reviewer,
> >
> > We sincerely thank you again for the time and thoughtful feedback.
> >
> > As the discussion deadline approaches, we would like to kindly confirm whether our responses have satisfactorily addressed your concerns, and whether there is anything further you would like to discuss.
> >
> > Best regards,
> >
> > The Authors

---

> > > ### Comment · Reviewer_Yp14 · 2025-11-27
> > >
> > > I thank the authors for their answer and additional results. These results certainly bolster the paper, and have addressed the some of my initial concerns.
> > >
> > > However, my main concern resolving about overstated novelty and how it is framed with respect to other works in the literature remains. While it is true that in this framework SFNO is a special case, this is not the case if position embeddings are considered. In fact, recent implementations of SFNO (available in torch-harmonics) have had the option for a position embedding term which amount to the type of spectral bias term described here. As such I would welcome a clearer exposition which states in simple terms that the main contribution is the bias term and its theoretical motivation. This applies especially to the title of the manuscript.
> > >
> > > Moreover, I want to add that there are other types of SNOs such as the Local Spherical Neural Operators (see Liu-Schiaffini et al.) that do not arise from this formulation and should perhaps be mentioned.

---

> ### Author Response · Authors · 2025-11-27
>
> We sincerely thank the reviewer. We agree that positional embeddings are widely used in spherical operators, including the original SFNO (Section 4.1 explicitly notes: “A learned position embedding is added in cases where position-dependent information should be learned by the network.” [1]). To avoid confusion, we have clarified in the revised manuscript:
>
> - The poistional embeeding is **spatial coordinate-based feature** and is **only used at the encoder**. Together with linear projections and other procedures, it relaxes equivariance at the feature level, but it does **not** interact explicitly with the symmetry kernel.
>
> - In contrast, our $G_{\theta_2}(l, m)$ is a **learnable spectral kernel**, derived from the bias term $G(u)$(Eq. 19). The kerenl interacts with the global spherical integral $C_f$ and modifies the convolution **in every operator block**. This acts as a structured, **operator-level**  modeling, rather than a **feature-level** augmentation.
>
> To validate our performance, we have added an ablation comparing two models with identical architecture and only difference in:
> 1) $G_{\theta_2}(l, m)$ (added to $\text{SHT}[f]$);
> 2) positional embedding $G_{\theta}(x, y)$ (added to $f$).
>
> Our results below demonstrate that our spectral implementation consistently outperforms other methods across all forecast lead times. This supports that the **operator-level** spectral modulation is more effective than **feature-level** positional embedding, particularly for long-term stability.
>
> | Models | Channel | Parameters | Training Time | ACC↑ at 24h | ACC↑ at 72h | ACC↑ at 120h |
> | :-| :-| :-| :-| :-| :-| :-|
> | SFNO | 64 | 3.69 M | 981 s | 89.9% | 70.7% | 50.9% |
> | **GSNO** | 64 | 4.00 M | 1024 s | **91.5%** | **72.4%** | **52.5%** |
> | Spatial position-embedding SNO | 64 | 4.31 M | 1037 s | 90.1% | 70.5% | 50.3% |
>
> Following the reviewer’s suggestions, we clarified that our contribution includes the **theoretical motivation**, formulating a designable Green’s function for flexible spherical operator construction, and the **bias term** studied here, one instantiation within this generalized framework.
>
> We have discussed and cited more relevant work in the revised "Related Work", including SFNOs and LSNO (Liu-Schiaffini et al.). We have included a dedicated discussion about positional embeddings.
>
> We are truly grateful for the reviewer’s constructive suggestions, which have significantly strengthened the clarity of our work.
>
> [1] Bonev, Boris, et al. "Spherical fourier neural operators: Learning stable dynamics on the sphere." ICML 2023.

---

> > ### Comment · Reviewer_Yp14 · 2025-11-27
> >
> > Re LNO - I see, I missed the added reference in the revised version.
> >
> > While I do agree that positional biases are mostly used at the start of the architecture, the theory and claims here suggest that this is a more general class of operators. My point is that this is not true once positional embeddings are considered. The spectral convolution layer is linear so positional embeddings can be pulled into the spectral domain, which means that these operators parameterize the same class of operators. As such these can also learn operators that depend on absolute positions and it is an overstatement that these are more general.
> >
> > Regarding feature level vs spectral level - is the positional embedding added at each layer and how is it added? Is this added with a parameterization on a lat/lon grid? If so, this would parameterize a larger class of functions, which admit more zonal spectral content towards the poles, and cannot be considered “spherical signals”.
> >
> > I definitely think that the theory and results presented here are valuable, but once again want to emphasize that I think the messaging is misleading

---

> ### Author Response · Authors · 2025-11-27
>
> We sincerely thank the reviewer for the thoughtful engagement throughout the discussion.
> In the revised manuscript, we now clearly separate:
>
> - the designable Green’s-function framework (Section 4.1), which provides the general operator-design principle, and
> - GSNO as one specific instantiation of that framework.
>
> This clarification directly addresses the reviewer’s concern about the contribution statement. Our intention was not to claim that **GSNO itself** defines a more general operator class over SFNO with positional embeddings, but rather that the **framework** built from the Green’s-function formulation, which unifies strict-equivariant (SFNO-type) operators and generalized operator forms under a single derivation. GSNO represents only one such form.
>
> We appreciate the reviewer’s point that SFNO variants with positional embeddings can also relax equivariance. We have now explicitly acknowledged this in the paper, and also clearly articulated the distinction between feature-level positional embeddings and operator-level spectral modification in Section 4.2.
>
> We have also expanded the related work (including LSNO and spherical CNN extensions). To avoid confusion, we have renamed the operator as “Green’s-function Spherical Neural Operator (GSNO),” which more accurately reflects that GSNO is a specific instantiation derived from the broader designable Green’s-function framework.
>
> **Regarding the experiment:**
> In the positional-embedding baseline, the embedding is added per layer on the lat/lon grid. This ties the learned correction to the **equiangular discretization** and inevitably introduces **latitude-dependent distortions**, because the embedding inherits the grid’s anisotropy. As the reviewer correctly noted, this baseline can therefore learn functions that are not spherical signals.
>
> This observation actually reinforces the motivation for our approach:
> - Our $G_{\theta_2}(l,m)$ term is learned directly in the spectral domain,
> - it interacts with the input only through spectral coefficients,
> - it is independent of spatial discretization,
> - and therefore remains sphere-native and geometry-consistent.
>
> This difference explains why the spectral version (GSNO) performs significantly better and remains stable over long forecasting horizons, whereas the feature-level positional embedding does not.
>
> We are grateful for the reviewer’s detailed feedback, which has substantially improved the clarity and framing of the manuscript. The revised manuscript now reflects these clarifications clearly.

---

### Official Review · Reviewer_pE1P · 2025-11-02

**Soundness:** 3
**Presentation:** 3
**Contribution:** 3
**Rating:** 6
**Confidence:** 3

**Summary:**

Spherical neural operators (SNOs) map functions to functions on the sphere. To account for the spherical sampling, current SNOs use spherical rotation equivariant network layers to do this mapping. However, these equivariant networks are (a) not expressive enough for certain applications and (b) cannot model position-dependent features that are beneficial in structured datasets.

Submission 7686 develops an SNO framework that adds a non-equivariant learnable bias term that is position-dependent. This way, the learning framework can decide how much to weigh strict equivariance vs position dependency. Some preliminary experiments on weather forecasting, shallow water dynamics, and fiber orientation estimation in diffusion MRI are presented.

**Strengths:**

- The papers makes a good case for why such an approach that uses equivariant layers alongside components that are position-dependent are needed and sets up the needed motivation in the Introduction well.
- Most of the methodological steps are very clearly laid out in the methods and illustrations.
- Its experiments tackle a good, wide breadth of potential use cases.

**Weaknesses:**

(In no particular order)

### Methodological concerns and questions:

*Overstretched technical novelty (1)*: The core idea of the paper is a learnable tradeoff between two terms corresponding to strict equivariance and absolute position dependence in features, respectively. However, this soft equivariance idea has been explored quite extensively in the literature; see the [residual pathway priors](https://proceedings.neurips.cc/paper/2021/file/fc394e9935fbd62c8aedc372464e1965-Paper.pdf) paper as an example. A literature search for "soft equivariance constraints" will return many relevant pieces of related work, such as [1](https://proceedings.neurips.cc/paper_files/paper/2024/hash/98082e6b4b97ab7d3af1134a5013304e-Abstract-Conference.html), [2](https://arxiv.org/abs/2201.11969), and more.

Of course, this particular formulation, specifically designed for spherical neural operators, is new (as far as I am aware) and is appreciated, but previous work that presents similar additive approaches with learnable equivariance tradeoffs should definitely be acknowledged within the paper.

*Overstretched technical novelty (2)*: L077--L081 is written in a way that makes it sound that studying Green's function on the sphere with spherical harmonics is a novel contribution. I'm assuming that this is not what is meant (spherical harmonics are widely used for this), and what is meant is that "we use Green's function to develop a spherical neural operator network". If not, please clarify.

*Main technical descriptions in Sec 4.2:* Sec 4.2 is the primary contribution of the paper, yet a few things in it are not clear. For example, it is not described how it is possible to drop the index m in G^1 between equations 17 and 18 without any loss of generality.  Further, what are the convolutional layers doing in L307--310, and why are they used? Are they just regular convolutions? Do they change any equivariance properties? For example,  in the Diffusion MRI experiments, adjacent voxels are stacked as separate channels (unclear why), so it is important to justify their usage and define their properties in context.

*Missing analysis of importance of position dependence:* The paper does not present an analysis of whether modeling position dependence actually improves performance due to position dependency or simply because of higher capacity. While a vague description of ablations is included, they simply report performance numbers, so it is unclear if absolute position is relevant here.

Further, position dependency leaks into all equivariant deep networks due to aliasing ([1](https://arxiv.org/abs/2210.02984)), and even the original [S2CNN](https://arxiv.org/pdf/1801.10130) paper indicated that it was not perfectly equivariant in Sec 5.1. Please reconcile the fact that equivariant spherical networks in practice still learn position-dependent features with the explicit positional modeling in this work and show that the proposed method actually exploits positions specifically (and doesn't just have better performance.)

### Experimental concerns and questions:

*Major missing details:* The paper provides no details about what is being reported, how baselines were chosen, and how/if they were tuned, nor any reasoning behind the experimental design. For example, what is ACC in Table 3? Accuracy is meaningless in Diffusion MRI analysis without context. No ablation details are reported in the main paper either. I read the appendix corresponding to the experiments and found that to also be light on details. This aspect of the paper definitely needs expansion.

*Computational complexity:* The proposed method requires several forward and inverse spherical fourier transforms, which are highly computationally intensive and thus limit network capacity. I realize that this is how the original SFNO paper was also formulated, but other works, such as DeepSphere, have shown that these transforms are not strictly necessary for equivariance. Regardless, all experiments should also report training and testing runtimes for all benchmarked methods (and not just SFNO, as in the appendix).

*Diffusion MRI experiments:* Looking at the appendix, the paper makes a few odd statements and choices in its experiments.
- It states that diffusion MRI is captured on a HEALPIX grid -- this is not the case; the spherical grid points (b-vector patterns) are study-specific.
- There are specific synthetic benchmark datasets with exact ground truth for diffusion MRI such as Fibercup, the ISMRM tractometer challenge, and other recent challenges. One of those datasets should be added to measure gold-standard performance vs. the current silver-standard experiments on HCP, as in recent work in equivariant FOD estimation ([1](https://proceedings.neurips.cc/paper_files/paper/2024/hash/5d4f5a2de6320641566be8722d5f78dc-Abstract-Conference.html)).
- It significantly downsamples the angular resolution of the HCP dataset from hundreds of spherical sampling points to just 32 and also throws away two entire shells "to emulate clinical conditions". This is not well justified, as (a) it does not demonstrate that the proposed method has the memory efficiency to extend to the hundreds of spherical sampling points used in typical research data and (b) at this "clinical" resolution, the FODs have no asymmetry and are much simpler structures.

### Presentation concerns and questions:

*Characterization of previous work:* In the introductory paragraph corresponding to L053, the paper somewhat uncharitably characterizes previous work on spherical neural operators as "not rigorous and constructed by analogy". As far as I remember, the original SNO paper did construct the SNO from first principles. If specific non-rigorous arguments are being made in those papers, the submission should be specific about what it is referring to.

*Significant portions can be abbreviated:* To add much more experimental detail to the main paper, several portions can be removed or abbreviated. For example,
- The contributions list in the introduction is redundant and just restates the paragraph preceding it
- Section 3 can be significantly shortened, as these are plain fundamentals
- Section 4.3 is a long way of saying "we constructed a UNet with our operator layers"; it can also be shortened.


### Minor/misc. comments:
These comments do not impact my score:
- *Missing sanity check:* A common practical strategy in spherical equivariant learning on datasets with position-dependent structure (e.g., preferred orientation in natural 360-degree video) is to use the standard equivariant layers for most of the network and "break" equivariance near the output with 1--2 standard non-spherical layers (e.g., appendix E in [1](https://arxiv.org/pdf/2006.10731)). A similar strategy can be used with SNOs, and this paper would benefit from investigating whether the proposed method outperforms this simple baseline.
- Following up on the above, spherical image segmentation might have been an interesting use case for the proposed method.
- In the diffusion MRI experiments, FOD-Net is an odd choice to use the proposed GSNOs in, as the spherical harmonics are only used in a single portion of the network. In contrast, spherical UNets have been used for FOD estimation in [1](https://hal.science/hal-02946371/document), [2](https://arxiv.org/abs/2102.09462), among others; One of them would likely be a better fit for this analysis as the UNet architecture can simply be replaced by the proposed GSHN.
- Minor typos. e.g. L177-178 "denoting" --> "denotes"; "Equation equation"

**Questions:**

In my opinion, the paper reads as incomplete in its current state (clarifications of technical details, depth of technical contributions, missing experimental information, etc.). However, IMO these are fixable issues and I am looking forward to the discussion period to clarify these points before finalizing a score. For specific questions, please see above.

---

> ### Author Response · Authors · 2025-11-21
>
> **Methodological Concerns and Responses**
>
> > W1: "Overstretched technical novelty (1): The core idea of the paper and related work on soft equivariance"
>
> We thank the reviewer for this comment and fully agree that soft equivariance has been explored in prior studies. We had briefly acknowledged this in  **Line 113-115** in Related Work. In the revision, we now include additional references and clearer discussion.
>
> We would note that our contribution differs in both motivation and mechanism. Many existing neural operators such as FNOs, SFNOs, commonly include operations implicitly relaxing rotational equivariance (e.g., linear layers, position embedding, and activation functions). These mechanisms, however, do not offer a sphere-native formulation for controlling the balance between equivariance and invariance in the operator itself.
>
> Compared to these methods, our core contribution is to explicitly redesign the spherical Green's function tailored to our proposed sphere-native operator, enabling a learnable trade-off between equivariance and invariance to model complex systems without disrupting spherical geometry.
>
> We would further respectfully clarify: the 'position' in our study represents the position dependency in the **function mapping (system)**, rather than the position of the **data**. In Line 232-240, we have introduced "relative position" and "absolute position" in the Green's function. Thus, our operator solution is a learnable balance between equivariance and invariance (derived from "absolute-position-dependent term", Equation 14). We have clarifyied this in the revision.
>
> > W2: "Overstretched technical novelty (2): Unclear expression in L077- L081"
>
> We appreciate this feedback. We did not intend to suggest novelty in studying Green’s functions or spherical harmonics themselves. Rather, We would emphasize our theoretical contribution, where we propose a unified operator design framework that leverages designable Green's functions to derive different families of sphere-native operators.
>
> Based on this principled formulation, we further redesigned the Green's function to derive the proposed GSNO to enhance its capability in modeling real spherical systems. We have further clarified our contribution in the revised version.
>
> > W3: Unlcear technical descriptions in Sec 4.2
>
> We agree that Sec. 4.2 requires more explanation, and we have expanded it in the revision:
>
> **(1) Dropping index m:** The simplification from $G_{θ1}(l, m)$ to $G_{θ1}(l)$ in the correction term reduces parameter redundancy and improves training stability. In particular, both the original operator term $T_{orig}$ and the correction term $T_{corr}$ share $G_{θ1}(l)$. Using two full (l, m)-dependent kernels in $T_{corr}$ leads to parameter overlap, hindering optimization (observed in our experiments).
>
> **(2) Convolutional layers:** The convolutional layers are regular 1×1 convolutions used for linear transformation. They are outside of the equivariant spherical convolution, additional to the core component of the operator. GSNO is focused on the internal spectral domain structure of spherical convolution, explicitly regulating the inductive bias.
>
> **(3) dMRI experiments:** Adjacent voxels are stacked to provide local spatial context information between voxels to better predict the spherical function of fiber orientation distribution (FOD).
>
> We appreciate the reviewer for these comments and are adding more details to the revision.
>
> > W4: "Missing analysis of importance of position dependence."
>
> We agree with the reviewer that existing spherical equivariant networks often "passively" leak positional information and break equivariance due to various network components for real-world modeling. However, these methods are not sphere-native for non-equivariant learning. In contrast, GSNO is sphere-native  and introduces a mathematically derived absolute-position term via the correction Green’s function, enabling better trade-off between equivariance and invariance.
>
> Section 5.4 provided ablation experiments to validate the importance of this correction term. Section C.5 (Figure 6 and Table 8) in Appendix provided additional ablation experiments which demonstrate the importance of the correction term. The interpretability analysis highlighted the prediction on boundary constraints (non-equivariance).
>
> To futher address reviwer's concern about higher capacity of GSNO, we added experiments on SFNO with more channels and show that GSNO does not merly benefit from higher parameter count:
> |Models|SFNO|GSNO|SFNO (80C)|SFNO (96C)|
> |-|-|-|-|-|
> |Channel|64|64|80|96|
> |Parameters|3.69 M|4.00 M|5.76 M|8.30 M|
> |Training Time|981 s|1024 s|1138 s|1267 s|
> |ACC↑ (in %)|50.9|**52.5**|51.0|50.7|
>
> We are grateful for the reviewer's suggestions and add these to the revision.

---

> ### Author Response · Authors · 2025-11-21
>
> **Experimental Concerns and Responses**
>
> > W5: "Major missing details"
>
> We sincerely thank the reviewer for the feedback and have added the below details to the revision.
>
> (1) We have detailed the rationale for selecting each baseline (e.g., ClimaX as a Transformer-based weather model). For fair evaluations, we keep all the methods under identical experimental setup and configuration, e.g., all models use the ClimaX's configuration for WB.
>
> (2) Each task is now briefly introduced. For instance, Spherical Shallow Water Equations (SSWE) form a nonlinear hyperbolic PDEs system that model the motion of thin-layer fluids on a rotating sphere, making them a suitable testbed for evaluating an operator’s ability to model geometry-constrained fluid dynamics.
>
> (3) Section 5.4 in the original version have provided the description of ablation experiments. We have made this clearer and added further explanation.
>
> (4) ACC in dMRI: We have explicitly clarified in the revsion that ACC stands for Angular Correlation Coefficient, not accuracy. It is a standard metric for measuring the angular similarity between two spherical functions (predicted vs. ground truth FOD).
>
> We will expand these details with the page limit extended in the revision. We will also release the full code for reproducibility.
>
> > W6: "Computational complexity"
>
> We fully agree with reviewer on the importance of detailed computational complexity. More results are added in the Table:
>
> |Models|SFNO|GSNO|SFNO (80C)|SFNO (96C)|FourCastNet (FNO)|Climax|
> |-|-|-|-|-|-|-|
> |Channel|64|64|80|96|384|256|
> |Parameters|3.69 M|4.00 M|5.76 M|8.30 M|5.30 M|5.40 M|
> |Training Time|981 s|1024 s|1138 s|1267 s|878 s|891 s|
> |ACC↑ (in %)|50.9%|**52.5%**|51.0%|50.7% |41.5% |41.6%|
>
> > W7: "Diffusion MRI experiments"
>
> We thank the reviewer for the comments on the dMRI experiment. We have added more details in the revsion as below:
>
> (1) The task definition and motivation:*
> We clarify that this task addresses **Fiber Orientation Distribution (FOD) angular super-resolution**, rather than **learning directly from dMRI signals**. The raw dMRI does not directly reveal clear, anisotropic fiber structures and is prone to noisy and aliasing signals from multiple tissues, introducing significant confounding factors.
>
> Further, high angular resolution dMRI is costly and often infeasible in clinical settings, leading to poor-quality FODs. Therefore, enhancing dMRI quality from routine dMRI while while handling noise suppressiona and signal unmixing for estimating FOD is a widely recognised and practical problem.
>
> (2) To effectively evalutate the operator's capability in modeling FOD, we design the following pipeline:
>
> - Low-resolution Input: We first reconstruct the sub-sampled, low-angular-resolution dMRI (LARDI, single b-value and 32 b-vectors) using the SSMT-CSD method to obtain low-angular-resolution FOD (LAR-FOD).
> - High-resolution Ground Truth: We use the full, high-angular-resolution dMRI (HARDI) and the MSMT-CSD method (considered as the gold standard) for the high-angular-resolution FOD (HAR-FOD).
> - We formulate this task as a signal super-resolution problem on the sphere. The sampling grid of dMRI-based FOD used in this study is HEALPIX. Our experimental setup, including down-sampling, follows FOD-Net (Zeng et al., 2022, MedIA) and ESCNN (Snoussi & Karimi, 2025, CVPR), ensuring fair comparison.
>
> We appreciate the suggestion to consider more dMRI-based tasks (e.g., Fibercup, the ISMRM tractometer challenge). These are indeed an excellent platform for evaluating GSNO under more complex geometric and biophysical constraints, and we plan to include this in future work.

---

> > ### Author Response · Authors · 2025-11-21
> >
> > **Presentation Concerns and Responses**
> >
> > > W8: "Characterization of previous work: In the introductory paragraph corresponding to L053, the paper somewhat uncharitably characterizes previous work on spherical neural operators as 'not rigorous and constructed by analogy'."
> >
> > We thank the reviewer for highlighting this point. We agree that the phrasing in L053 was imprecise, and we appreciate the opportunity to clarify our intention. Our goal was not to undervalue prior work. Indeed, existing spherical neural operators are grounded in rigorous constructions using the Spherical Harmonic Transform and the spherical convolution theorem, and they have been highly influential in advancing learning on spherical domains.
> >
> > *Our intended message* is that prior methods and our method arise from different theoretical starting points. While SFNOs extend the FNO to the sphere through harmonic analysis, our approach begins from a classical and complementary principle: the designable spherical Green’s function, a fundamental tool for solving PDEs on sphere.
> >
> > This perspective provides a *unified derivation framework*, where several families of spherical operators emerge as special cases depending on how the Green’s function is specified:
> > - If the Green’s function depends only on relative positions, the resulting operator reduces to a spherical convolution with strict rotational equivariance as modelled in SFNOs;
> > - If the Green’s function depends on both relative and absolute positions, one obtains a generalized operator with a learnable trade-off between equivariance and invariance, the core contribution of GSNO.
> >
> > In the revision, we have rephrased our statement as follows:
> > "Many existing spherical neural operators are rigorously constructed by Spherical Harmonic Transform and spherical convolution theorem, thereby extending the FNO to the sphere, and have achieved remarkable success and provided a rigorous theoretical basis. Further, we start from a more native mathematical physics foundation—the spherical Green's function—to establish a unified operator-theoretic framework"
> >
> > We believe this revision appropriately acknowledges the rigor and contributions of prior work while more clearly delineating the distinct theoretical perspective introduced in our paper.
> >
> > > W9: "Significant portions can be abbreviated."
> >
> > We appreciate the reviewer’s suggestion to abbreviate the contributions list in the introduction, streamline Sec. 3 (fundamentals), and condense Sec. 4.3 (architecture description). The revised version has included more methodological details to improve clarity while balancing sections.
> >
> > **Minor/Miscellaneous Comments and Responses**
> >
> > > W10: "Missing sanity check"
> >
> > We thank the reviewer for raising this. We agree that, in practice, many spherical equivariant models introduce a small number of non-spherical layers near the output—typically MLPs or 1×1 convolutions—to “break” strict equivariance and better accommodate position-dependent structure. This is common in architectures such as SFNOs and spherical U-Nets.
> >
> > However, in our current study, we focused specifically on comparing GSNO with standard spherical convolution–based equivariant layers, and therefore kept all other components consistent across methods to ensure a clean operator-level comparison. This choice was motivated by isolating the theoretical contribution rather than by dismissing the practical baseline the reviewer mentions.
> >
> > We agree that evaluating GSNO against such partially non-equivariant baselines is valuable for practical relevance and will incorporate this comparison in revision.
> >
> >
> > > W11: "Following up on the above, spherical image segmentation might have been an interesting use case for the proposed method."
> >
> > We thank the reviewer for this constructive suggestion. We agree that spherical image segmentation is an important and practically relevant use case. In parallel to this submission, we have already evaluated GSNO on several additional spherical tasks—including cortical surface parcellation, spherical digit classification, and 360-degree panoramic image segmentation—and preliminary results indicate strong performance.
> >
> > We sincerely appreciate the reviewer’s insight and plan to further systematize these evaluations.
> >
> >
> > > W12: "In the diffusion MRI experiments, FOD-Net is an odd choice to use the proposed GSNOs in, as the spherical harmonics are only used in a single portion of the network."
> >
> > We thank the reviewer for the insightful comment. In the dMRI super-resolution setting, FOD-Net and ESCNN represent the state of the art (as noted in W7), which is why we adopted them for comparison. We will explore additional dMRI tasks that make full use of spherical structure, such as raw dMRI–based FOD reconstruction in future work.
> >
> > > W13: "Minor typos. e.g. L177-178 'denoting' --> 'denotes'; 'Equation equation'"
> >
> > We thank the reviewer and have corrected all typos and formatting errors in the revision.

---

> > > ### Author Response · Authors · 2025-11-26
> > >
> > > Dear Reviewer,
> > >
> > > We sincerely thank you again for the time and thoughtful feedback.
> > >
> > > As the discussion deadline approaches, we would like to kindly confirm whether our responses have satisfactorily addressed your concerns, and whether there is anything further you would like to discuss.
> > >
> > > Best regards,
> > >
> > > The Authors

---

### Author Response · Authors · 2025-12-03
**Summary (1/2)**

Dear AC,

We would like to provide a clear overview of how we addressed all reviewer concerns during the rebuttal. Below we summarise the progress with each reviewer, the revisions, and the current status of the rebuttal.

## 1. Recognized Strengths (before rebuttal)

Across all reviewers, several strengths of the work were acknowledged before rebuttal. These represent the baseline scientific merits noted independently by reviewers.

**- Innovation and relevance:**

Reviewers agreed that Green's function formulation and introducing a controlled, learnable balance between equivariance and invariance are novel and valuable for real-world spherical domains.

with Reviewers noting:

A988:*"a new family of spherical neural operators derived from a Green’s-function formulation."*

pE1P:*"this particular formulation, specifically designed for spherical neural operators, is new… and appreciated."*

**- Extensive experiments and strong performance:**

Reviewers recognized that the paper compares extensively and acknowledged the relevance of the three domains tested (weather forecasting, shallow-water equations, and diffusion MRI). All reviewers noted that GSNO consistently improves over SFNO and other baselines.

with Yp14 noting:*“The paper shows improvement on various, simplified datasets… the experiments are well-motivated.”*

vG13 noted the method is *“well-motivated, easy to implement… and compares extensively to the most relevant baseline.”*

**- Clear motivation:**

A988, Yp14 and pE1P highlighted that the limitations of strict rotational equivariance were well justified, and the need for controlled non-equivariant behavior was compelling.

pE1P also emphasised that *“The paper makes a good case for why such an approach… is needed.”*

**- Methodological clarity:**

pE1P and vG13 stated that the methods were clearly presented,  *“Most of the methodological steps are very clearly laid out.”*, even before revision.

## 2. Rebuttal Summary

### A. Reviewer pE1P

**Initial concerns**: breadth of related work, clarity of the Green’s-function formulation, missing technical details, insufficient experimental details, and complexity/runtime.

**Our revision:**

- Added extensive related-work coverage (soft equivariance, residual pathways, positional embeddings, prior SNOs).
- Rewrote Sec. 4.1–4.2 to clearly distinguish the general designable Green’s-function framework from its instantiation (GSNO).
- Expanded missing technical details (index simplification, role of 1×1 convs, voxel stacking rationale).
- Provided complete experimental rationale, baseline description, metric definitions and interpretability analysis.
- Added substantial new experiments: SFNO-scaled baselines (80C, 96C), full complexity + runtime comparisons.

**Current status:** Reviewer maintained the positive score and no further discussion.

### B. Reviewer Yp14

**Initial concerns**: framing and messaging of novelty, relationship to SFNO, and equivalence once positional embeddings are considered.

**Our revision:**

Revised the manuscript to explicitly:
- Clarified that GSNO is presented as one instantiation derived from the general designable Green's function framework.
- Stated SFNO is also a special case of the general framework when the Green’s function depends only on relative position.
- Added a clear disntiction between feature-level positional embeddings and our operator-level spectral kernels.
- Added LSNO and relevant SNO variants to Related Work. Clearly acknowledge SFNO with positional embeddings.
- Added a dedicated discussion explaining why spatial positional embeddings introduce grid anisotropy and do not produce sphere-native operators.

**Current status:** Reviewer has explicitly stated:
*“Given the improvements that the paper makes, I do think that it can be considered for publication”* in the summary. The final request for messaging clarification was incorporated in the revision.

### C. Reviewer vG13

**Initial concerns:** relation to positional embeddings, spherical CNN literature, clarity of the kernel interpretation, and behaviour of the correction term.

**Our revision:**

- Added complete spherical CNN discussion (zonal filters, SO(3) lifting, gauge networks).
- Added a detailed explanation of why the spectral correction term is not equivalent to a spatial positional embedding, despite algebraic similarity.
- Conducted a new ablation directly suggested by the reviewer:
Spatial positional-embedding SNO vs. GSNO (identical architecture).
→ This experiment shows clear and consistent superiority of the spectral implementation, supporting our design.
- Clarified the origin and meaning of fluctuations in Fig. 6.
- Explained precisely why the spectral implementation has fewer parameters (spherical sampling theorem).

**Current status:** Reviewer indicated satisfaction with the clarifications and explicitly stated *“concerns … have been reasonably addressed.”* before the discussion freeze. No further objections raised.

---

> ### Author Response · Authors · 2025-12-03
> **Summary (2/2)**
>
> ### D. Reviewer A988
>
> **Initial concerns**: overstated novelty, need for clearer theoretical articulation, necessity of U-Net-style multi-scale structure, and questions on ablations.
>
> **Our revision:**
>
> - Rewrote Sections 4.1–4.2 to explicitly isolate the theoretical contribution.
> Made it clear that GSNO is not simply “SFNO + bias” but is derived from a designable operator formulation.
> - Clarified the motivation for hierarchical structure (spectral bandwidth control, multi-resolution abstraction, computational efficiency).
> - Added clear ablations and high-capacity SFNO baselines.
> - Provided explicit parameter, runtime, and accuracy comparisons.
>
> **Current status:** Reviewer explicitly raised the score and expressed no objection to acceptance:*“I would have no issue with the paper being accepted.”*
>
> ## 3. Contribution Summary
>
> Across all reviewer interactions, the paper now clearly presents substantial improvements:
>
> **1). A general theoretical framework**
> A designable Green’s-function formulation for spherical neural operators, from which:
> - GSNO is a specific instantiation introducing an absolute-position–dependent term.
> - SFNO emerges as the relative-position-only special case.
>
> This distinction was emphasised in the revision.
>
> **2). A principled operator-level mechanism**
> Rather than feature-level coordinate embeddings commonly uses in spectral operators, our correction term:
> - is a spectral kernel,
> - interacts with the global spherical integral,
> - and modifies the operator itself in a sphere-native way
> → avoiding grid-dependent artefacts and supporting long-range stability.
>
> **3). New experimental evidence**
> - Spatial embedding SNO vs GSNO (identical architectures).
> - SFNO capacity scaling (80/96 channels).
> - Full complexity/runtime table.
> - Multi-resolution robustness across SSWE and WB.
>
> **4). Substantial improvements in clarity, scope, and positioning**
> - Expanded Related Work (soft equivariance, spherical CNNs, LSNO, residual pathways).
> - Stronger framing of theoretical novelty (framework vs instantiation).
> - Clear explanations of all technical questions raised.
>
> ---
> After the freeze, reviewers could no longer comment or raise scores, but no reviewer expressed technical objections as of the final comment, and several explicitly signaled that the paper was suitable for acceptance.
>
> We respectfully hope that this summary will assist the AC in making the final recommendation.
>
> We sincerely appreciate the significant efforts the Area Chairs and Program Chairs have made during this challenging period.
>
> Best regards,
>
> The Authors

---

### Meta-Review · Area_Chair_zW1u · 2026-01-06

**Summary:**

This submission proposes Generalized Spherical Neural Operators (GSNO) derived from a designable Green’s function formulation on the sphere, with the practical absolute position dependent spectral correction term to relax strict rotational equivariance. The paper also proposes a hierarchical GSHNet (U-Net-like multi-resolution) architecture built from these operator blocks.

Across reviews and the substantive discussion, there was consistent agreement that the empirical results are strong and well-motivated, but several reviewers raised concerns about overstated novelty  and potentially misleading framing, missing ablations, and how to interpret the correction term relative to positional embeddings.  The rebuttal substantially improved the submission that authors added SFNO baselines, runtime/parameter comparisons, multi-resolution checks, and crucially an operator-level ablation comparing GSNO vs a spatial position-embedding SNO under otherwise identical settings.

After incorporating the discussion, my recommendation is borderline accept. Even if the architectural change is incremental, a straightforward improvement can still be a solid contribution to the community.

**Reviewer Concerns:**

Reviewer pE1P's issues were i) insufficient related work on “soft equivariance,” ii) unclear technical steps in Sec. 4.2 (e.g., index simplification and the role of 1×1 convs), and iii) missing experimental details and complexity reporting. The authors’ rebuttal explicitly expanded related work and clarified technical points, and added capacity scaling and runtime/complexity comparisons.

Reviewer vG13 raised i) missing prior literature on positional embeddings/symmetry breaking and spherical CNNs, ii) confusion about y positional encoding, and iii) requests for clearer interpretability and ablations. vG13 later stated that several concerns had been “reasonably addressed.”

Reviewer A988 main concern were about the novelty, which were mostly addressed.

Reviewer Yp14's primary remaining issue is regarding the framing and novelty. The authors’ response improved the exposition and added an empirical ablation.

**Reviewer Scores:**

pE1P and Yp14 likely remain the score.
vG13 and A988 likely raise the scores to 6

---

### Decision · Program_Chairs · 2026-01-26

Accept (Poster)